

# Inferring the destabilization susceptibility of mountain permafrost in the French Alps using an inventory of destabilized rock glaciers

Marco Marcer[1,2], Charlie Serrano[1,2], Alexander Brenning[3], Xavier Bodin[2], Jason Goetz[3], and Philippe Schoeneich[1]

[1]Institut d'Urbanisme et Géographie Alpine, Université Grenoble Alpes, Grenoble, France
[2]Laboratoire EDYTEM, Centre National de la Recherche Scientifique, Université Savoie Mont Blanc,Le Bourget-du-Lac, France
[3]Department of Geography, Friedrich Schiller University Jena, Jena, Germany

**Correspondence:** Marco Marcer (marco.marcer@univ-grenoble-alpes.fr)

**Abstract.** Knowing the extent of degrading permafrost is a key issue in the context of emerging risks linked to climate change. In the present study we propose a methodology to estimate the spatial distribution of this phenomenon, focusing on the French Alps. At first, using recent orthoimages (2000 to 2013) covering the study region, we mapped the geomorphological features that can be typically found in cases of rock glacier destabilization (e.g. crevasses and scarps). This database was then used as support tool to rate rock glaciers destabilization. The destabilization rating was assigned also taking into account the surface deformation patterns of the rock glacier, observable by comparing the orthoimages. The destabilization rating served as database to model the occurrence of destabilization in relation to terrain attributes and to predict the susceptibility to destabilization at the regional scale. Potential destabilization could be observed in 58 rock glaciers, i.e. 12% of the total active rock glaciers in the region. Potentially destabilized rock glaciers were found to be more prone to strong acceleration than stable rock glaciers within the period 2000 - 2013. Modelling the occurrence of destabilization suggested that this phenomenon is more likely to occur in elevations around the 0°C isotherm (2700 – 2900 m.s.l.), on north-exposed, steep (up to 30°) and flat to slightly convex topographies. Model performances were good (AUROC: 0.76) and the susceptibility map reproduced well the observable patterns. About 3 km$^2$ of creeping permafrost, i.e. 10 % of the surface occupied by active rock glaciers, had a high susceptibility to destabilization. Only half of this surface is currently showing destabilization evidence, suggesting that a significant amount of rock glaciers are candidates for future destabilization.

## 1 Introduction

For the past decades permafrost in the European Alps has been showing signs of widespread degradation (Haeberli et al., 1993, 2010; Springman et al., 2013; Bodin et al., 2015). Warmer mean annual and extreme warm air temperatures are expected to eventually cause a shift of the lower limit of mountain permafrost by several hundred meters towards higher elevations in the near future (Hoelzle and Haeberli , 1995; Lambiel and Reynard , 2001). However, due to the thermal inertia of the soil, permafrost may persist for decades to centuries in climatic conditions currently not favourable to the existence of frozen ground (Scapozza et al., 2010). Active layer thickening (e.g. Hilbich et al., 2008), increase of the liquid water content in the frozen soil



matrix (e.g. Ikeda et al., 2008), and other processes cause frozen slope materials to lose cohesion (Haeberli et al., 1997; Harris and Davies , 2001; Nater et al., 2008; Huggel et al., 2010). Abnormal rockfall activity at high elevations (e.g. Ravanel and Deline , 2010), thermokarst formation (Kääb and Haeberli , 2001) and extensive destabilization of active rock glaciers (Roer et al., 2008; Delaloye et al., 2008; Serrano , 2017) are indicators of this change of state in the mountain permafrost. Since these

events represent a serious threat to alpine communities, there is a growing need to understand where permafrost destabilization is occurring at a regional scale to allow for better risk assessment and land use planning (Haeberli et al., 2010).

Despite the importance of this issue, there is still no well-established methodology for evaluating the extent of permafrost degradation in mountain environments. Thermal evolution of the frozen ground was successfully modelled using numerical approaches at the regional scale in the Arctic (Westermann et al., 2016). Although capable of excellent results, these methods

require data at high spatio-temporal resolution that may be challenging to acquire. In the European Alps, a few studies intended to evaluate the spatial footprint of degrading permafrost using a theoretical approach based on shifting permafrost maps by an elevation equivalent to the isotherm rise of the past century (Hoelzle and Haeberli , 1995; Lambiel and Reynard , 2001). This method suggests that permafrost degradation occurs on a latitudinal belt of about 200-300 m. However, empirical evidence doesn't seem to fully agree with this approach as, for instance, stable and unstable rock glaciers may be found at the same

altitude (Serrano , 2017). Also, Sattler et al. (2011) showed that permafrost initiation points of debris flows weakly correlate with the spatial footprint of degrading permafrost suggested by this methodology. Statistical approaches on the other hand, have shown promise for modelling the distribution of slope failures related to permafrost degradation. Following previous works on landslide susceptibility (e.g. Goetz et al., 2011), Rudy et al. (2017) illustrated how non-linear statistical modelling can be applied to spatially predict active-layer detachment failures in the Canadian Arctic in relationship to terrain characteristics.

In this context, observing rock glacier dynamics and morphology can be rather useful. Permafrost degradation in ice-rich landforms causes mobilization of significant amounts of materials that, in particular topographic settings, may trigger or pre-condition debris flows (Kummert and Delaloye , 2018; Schoeneich et al., 2017). An increase of the liquid water content of the landform to a critical point can cause the so-called destabilization (Ikeda et al., 2008; Roer et al., 2008; Scotti et al., 2016; Bodin et al., 2016). Destabilization is a dynamical condition that affects active rock glaciers and is characterized by high dis-

placement rates which may increase sharply in time in a "surge" type phenomenon (Delaloye et al., 2008). This state results in drastic morphological changes and may eventually lead to the collapse of the rock glacier causing landslides (Bodin et al., 2016). The destabilization process of active rock glaciers can be observed since it produces typical geomorphological features on the surface of the rock glacier, such as crevasses and scarps, similar to landslides (Kaufmann and Ladstädter , 2003; Avian et al., 2005; Roer et al., 2008; Delaloye et al., 2008; Lambiel and Reynard , 2001; Scotti et al., 2016). These geomorphological

features, which are referred to as "surface disturbances" in this study, can be observed and mapped from aerial imagery (Roer et al., 2008; Serrano , 2017). Therefore, given the availability of multi-temporal imagery, an inventory of active rock glacier surface disturbances may be obtained at a large scale and it can be used for regional modelling of destabilization occurrence.

The purpose of this study was to obtain regional-scale insights into the issue of destabilizing rock glaciers and degrading per-mafrost in the French Alps. In this region periglacial environment is abundant and occurrence of rock glacier degradation has

been observed (Echelard , 2014; Bodin et al., 2016; Serrano , 2017; Schoeneich et al., 2017). Periglacial hazards therefore exist



and, given the dense urbanization of this region, the need for tools allowing a comprehensive risk assessment is crucial (Bodin et al., 2015). To do so, surface disturbances on active rock glaciers were mapped by multi-temporal aerial image interpretation based on expert field knowledge. Surface disturbances were then used as support tool to assign a destabilization rating ranging from 0 to 3 to each active rock glacier. Rock glaciers classified with higher destabilization rating presented typical geomorpho-
logical characteristics reported in known cases of destabilization, as pronounced surface disturbances that increased by number and size in the past decades. These rock glaciers were suggested to be potentially destabilized and hypothesized as evidence for permafrost destabilization. On the other hand, rock glaciers not presenting surface disturbances were classified with lower ratings of destabilization (i.e. stable rock glaciers) and hypothesized as evidence for absence of permafrost destabilization. Horizontal displacements of the rock glaciers were also evaluated in order to compare destabilization rating and kinematics.
This was done by manually feature tracking moving boulders on the landforms surface.

Evidence of presence/absence of permafrost destabilization was used to model rock glacier stability in relation to terrain parameters by using a statistical approach similar to Rudy et al. (2017). The resulting susceptibility map provides an overview of potentially destabilizing permafrost areas at the regional scale. We refer at this map as the DEFROST (destabilizing permafrost) susceptibility map, which was finally used to propose a diagnostic of the periglacial environment in the region by
evaluating destabilization susceptibility of the active rock glaciers.

## 2   Methods

### 2.1   Study area and rock glacier inventory

The French Alps cover an area approximately 50-75 km wide and 250 km long, located between 44° and 46°S and 5.7° to 7.7°W. In this region, almost 15 000 km$^2$ are above 1500 m.s.l., and permafrost is estimated to cover up to 770 km$^2$ (Boeckli
et al., 2012; Marcer et al., 2017). Rock glaciers are a common feature with more than 2600 units inventoried, of which almost 500 classified as active (Marcer et al., 2017).

The 0°C annual isotherm ranges from 2700 m a.s.l. in the Southern ranges down to 2200 m a.s.l. in the Northern sectors (Durand et al., 2009). According to Auer et al. (2007),mean annual air temperature increased by up to 1.4°C in the whole Great Alpine Region during the 20th century, and this rate has been accelerating in recent decades. As result, permafrost temperatures
retrieved by borehole observations are suspected to increase at a rate of 0.04°C per decade (Schoeneich et al., 2012). The increase of rock glacier velocity and their destabilization observed in the region since the late 1990s is also suggested to be linked to this phenomenon. In 2006 the Berard rock glacier collapsed causing a landslide of 250 000 m$^3$ (Bodin et al., 2016). Echelard  (2014) identified another case of a striking destabilization, the Pierre Brune rock glacier, which was developing a series of deep crevasses while also accelerating. In 2015, the active layer of the frontal lobe of the Lou rock glacier detached,
causing a debris flow that flooded the town of Lanslevillard (Schoeneich et al., 2017). In a first attempt to get a regional overview, Serrano  (2017) mapped destabilized rock glaciers in the Maurienne valley, Vanoise national park and Ubaye valley, highlighting the high incidence of destabilized rock glaciers in these areas.





A rock glacier inventory of the French Alps belonging to L'Office national des forêts (ONF: the National Forest Office) was used in this study (Roudnitska et al., 2016; Marcer et al., 2017). The inventory was compiled between the years 2009 – 2016 by inspecting aerial imagery in Geographical Information System (GIS). Although activity was attributed by interpreting the morphologic attributes of the landforms, Marcer et al. (2017) noticed that observing multiple orthoimages taken at different

date can reduce the uncertainty in attributing the activity of the landforms. Therefore in this study, rock glaciers were classified as active if movements are observable in multi-temporal orthoimagery. Only active rock glaciers will be considered from now on.

## 2.2 Mapping rock glacier destabilization

The first step to identify destabilized rock glaciers was mapping surface disturbances on rock glaciers. Previous studies that

described destabilized rock glaciers showed that these landforms present a wide variety of geomorphological features (e.g. Roer et al., 2008). Here, we followed a methodology similar to Serrano (2017), which consisted of defining a catalogue of typical surface disturbances that can be found on destabilized rock glaciers. Surface disturbances on rock glaciers were classified in four distinct categories, depending on their morphology and triggering causes: debris flow gullies, cracks, crevasses and scarps (Figure 1, Table 1).

In this study, surface disturbances and movements were mapped for the inventoried rock glaciers based on interpretation of a set of multi-temporal high-resolution aerial imagery for the French Alps. This orthoimagery collection was obtained from the Institut géographique national (IGN, National Institute of Geography), which is freely available from the official website (www.geoportail.fr) or can be accessed as a Web Mapping Service (IGN , 2011a, 2013). The IGN orthoimagery collection consists of orthomosaics covering all of France for three different collection periods. The first orthomosaic is composed of

images taken from 2000 to 2004, the second from 2008 to 2009, and the third from 2012 to 2013. All images are of high-resolution: 2 m x 2 m for the two older mosaics, and 50 cm x 50 cm for the most recent mosaic.

Using a single orthoimage to map surface disturbances can lead to misinterpretations in the case of poor illumination of the terrain and snow pathces covering the ground (Serrano , 2017). Indeed, as the surface morphology of a rock glacier is naturally shaped according to spatially varying creep patterns, it is easy to mistake actual surface disturbances to compression features,

as furrows, depending on image quality. Therefore, surface disturbances, i.e. those morphological features not related to the creeping of the ice-rich permafrost, were mapped using all three available orthoimages in order to check that actual strain occurred where surface disturbances are located and to overcome limitations related to poor quality of an individual image.

### 2.2.1 Rating the degree of destabilization

After the rock glacier surface disturbances were mapped, a rating of the degree of destabilization was assigned to each rock

glacier. This rating was given not only to provide some insight to the observed levels of destabilization in the French Alps, but also to provide a confidence rating to describe a rock glacier as stable or unstable for the spatial distribution modelling of rock glacier destabilization.



Assigning a rating to quantify the degree of destabilization of a rock glacier required the definition of the characteristics of the "typical" destabilized rock glacier. Roer et al. (2008) suggested that rock glacier destabilization is observable when a sharp velocity increase and morphological disturbance occurs. They further related these changes to a shift in the underlying dynamical processes from creep towards basal sliding. Although we agree with this definition, in the present study we propose
a slightly different definition of destabilized rock glacier, determined by analysing historical aerial oblique photography and dynamical behaviour of the known cases of destabilization.

The milestone case of destabilized rock glacier in the French Alps, the Berard rock glacier showed a crevasse since 1947, which did not evolve until the early 2000s (Bodin et al., 2016). In 2003 the crevasse seemed to deepen and a new one formed a few tens of meters further east of the original. The rock glacier collapse took place where these crevasses were located. In the
Pierre Brune (Figure 2), Roc Noir and Hinteres Langtalkar rock glaciers a series of scarps and crevasses cut the whole body and divided the rock glacier into two zones with different velocities (Echelard , 2014; Serrano , 2017; Roer et al., 2008). Although surface disturbances could be observed in aerial imagery since the 1940s to the 1960s, their evolution in terms of quantity and size were linked to the increased displacement speed of the sectors of the rock glacier downstream the surface disturbance, which occurred since the 1990s. Earlier, the rock glacier seemingly creeped uniformly. A similar pattern was observed on the
Plator, Grosse Gralbe and Gander rock glaciers, where a scarp marked the sharp transition from displacement speeds in the order of 0.1 – 0.9 m/y to displacements speeds of the order of several meters per year (Scotti et al., 2016; Delaloye et al., 2008).

These observations suggested that the presence of surface disturbances was a necessary but not sufficient condition to the occurrence of destabilization, as rock glaciers may present surface disturbances but be stable for decades. Also, high speeds may not be a necessary feature, as some destabilized rock glaciers, e.g. Lou and Furggwanghorn, moved at a "normal" rate of
around 2 m/yr (Schoeneich et al., 2017; Roer et al., 2008). On the other hand, the agreement between the discontinuity of the surface deformation pattern of the rock glacier and the surface disturbances was suggested to be a key pattern in destabilization. The co-occurrence of these two conditions was found in every known case of destabilization here analysed. Considering this, we proposed a rock glacier destabilizing rating that varied from 0 (stable rock glaciers) to 3 (rock glaciers potentially destabilized, Table 2). For each active rock glacier, a rating of the degree of destabilization was assigned by observing the combination of
surface disturbances and a qualitative assessment of recent deformation patterns. This rating was applied using a standardized workflow (Figure 3). Temporal evolutions were assessed by observing the IGN orthoimagery collection.

### 2.3  Modelling rock glacier stability - the DEFROST index

The modelling followed a statistical approach similar to previous spatial prediction studies on landslides (Goetz et al., 2011) and arctic permafrost (Rudy et al., 2017) that used the Generalized Additive Model (GAM) with logistic link function (R
package "mgcv"). GAM was selected because of its flexibility in modelling non-linear interactions between dependent and predictor variables. The logistic link function allows to model the occurrence of a categorical response variable (response variable) as a function of continuous variables (predictor variables). In this study, rock glacier stability was hypothesized to be caused by a series of local morphological conditions. In particular, rock glacier destabilization grouped by either presence




or absence was the response variable, while terrain attributes describing local topography and climate were used as predictor variables.

Variables were sampled at the active rock glacier locations. At first, a point grid at the resolution of 25 m x 25 m was generated within the active rock glaciers polygons and used to sample the response and predictor variables values. Since the

rock glacier inventory counted a relatively small number of potentially destabilized cases (58 individuals), selecting only one point per rock glacier would have caused large uncertainty in the model outcome. Therefore, multiple points were randomly selected within each rock glacier perimeter. Since model performances were found to stabilize for more than five points selected per rock glacier, this number of points was randomly extracted per rock glacier for modelling.

The multiple variable models were computed using different combinations of predictor variables. Different models were

compared using the Akaike Information Criterion (AIC), which is a measure of goodness of fit that penalizes more complex models. The multiple variable model performing the lower AIC was selected to describe the occurrence of destabilization. The final multiple variable model was selected by iterating a backward-and-forward stepwise variable selection, aimed to identify which combination of predictors was better at describing the response variable by means of lower AIC

Model performance was estimated using the Area Under the Receiver Operating Characteristic (AUROC) (Hosmer and

Lemeshow , 2000). The AUROC estimates the ability of the model to discriminate stable and unstable areas. Sensitivity, i.e. the true positive rate, and specificity, i.e. the true negative rate, were used as additional criteria.

The predictive power of the model was estimated by spatial cross-validation (R package "sperrorest"). The method selected was the k-means clustering, which consisted in dividing the database in $k$ spatially contiguous clusters (Ruß and Brenning , 2010a). All but one clusters were used to train the model, while the remaining cluster was used to test the predictive power of

the model. This process was repeated until each cluster was used at least once in both training and test sets. Here, we divided the database into $k = 5$ clusters per run and used 100 repetitions. Performance indicators were evaluated on the respective test sets, and the overall model performance was evaluated using the average and standard deviation over all partitioning clusters.

The variable importance was assessed using permutation-based variable importance embedded in the spatial cross-validation (Ruß and Brenning , 2010b). This method consisted of permutating the values of each predictor variable one at a time and

calculating the reduction in model performance caused by the permutations. One thousand permutations were performed for each spatial cross-validation repetition. Predictor variables causing higher deviations while permutated were considered the most important ones in the model.

### 2.3.1   Model response variable

Surface disturbances were used as evidence of creeping permafrost destabilization. As surface disturbances were digitized

as linear features, they were buffered and merged into an "unstable areas" polygon database. A buffer distance of 30 m was chosen. The model was found to be insensitive to changes in buffer size up to 90 m. All remaining areas within the rock glacier polygons were used as "stable areas".

Polygons of both unstable and stable areas were sampled in order to assign the response variable to the modelling database. This was done under the hypothesis that surface disturbances were the geomorphological expression of destabilized permafrost.





However, many surface disturbances could be observed on rock glaciers that were classified as unlikely destabilized or as suspected of destabilization. On the other hand, in potentially destabilized rock glaciers surface disturbances could be observed to increase in time by number and size, creating a discontinuity in the deformation pattern, suggesting a stronger evidence of destabilization. Therefore, only unstable areas located in potentially destabilized rock glaciers were considered as solid

evidence of permafrost destabilization and assigned a destabilized permafrost (DEFROST) index value of 1. On the contrary, only stable areas belonging to stable and unlikely destabilized rock glaciers were used as sampling locations for evidence of absence of DEFROST and assigned a DEFROST index equal to 0. The DEFROST index values were then used as binary response variable with values of 0 for stable and 1 for potentially destabilized in the modelling stage.

### 2.3.2   Model predictor variables

Terrain attributes used in modelling were elevation, slope, profile curvature, potential incoming solar radiation and potentially thawing permafrost. This set of terrain attributes was selected aiming to represent the preconditions and processes causing the occurrence of destabilization in rock glaciers reported in previous studies. Rock glacier destabilization was observed to occur in rock glaciers at the lower limits of the permafrost zone in steep and convex slopes (Roer et al., 2008; Delaloye et al., 2008; Lambiel and Reynard , 2001; Bodin et al., 2016; Scotti et al., 2016).

Terrain attributes were derived from the BD Alti DEM, 25 m × 25 m spatial resolution (IGN , 2011a). Slope angle and downslope curvature (Freeman , 1991) were evaluated using the Morphometry Toolbox in SAGA GIS (version 2.2.2, Conrad et al. 2015). Negative values of curvature indicate concave topography, while positive values indicate convex topography. Also Potential Incoming Solar Radiation (PISR) was calculated using the Terrain analysis toolbox in SAGA as the sum of the computed direct and diffusive components of the radiation (Wilson and Gallant , 2000). Clear-sky conditions, a transmittance

of 70 %, and absence of a snow cover were assumed in the calculation of the annual total PISR.

The spatial distribution of potential permafrost thaw was evaluated using the analytical method already presented by others (Hoelzle and Haeberli , 1995; Lambiel and Reynard , 2001; Damm and Felder , 2013). The method consisted in artificially shifting a permafrost map proportionally to the estimated climate warming occurred between the period of validity of the map and the current climate. Here, we used the Permafrost Favourability Index (PFI) map (Marcer et al., 2017), which represents

the permafrost conditions during the cold episodes of the Holocene, e.g. Little Ice Age (LIA). The mean annual air temperature difference between the years 1850-1920 and 1995-2005 was determined using the HISTALP database (Auer et al., 2007) over the region. Temperature differences were then converted into equivalent elevation differences using the temperature lapse rates from Gottardi  (2009), and the PFI map was recomputed using the model parameters presented by Marcer et al. (2017). The resulting map, which corresponded to a theoretical permafrost distribution in equilibrium with the current climate, was finally

subtracted from the PFI, obtaining the Potential Thawing Permafrost zone (PTP, i.e. the so-called "melting area" in Lambiel and Reynard  (2001)). Since the PTP is a difference between favourability indexes, ranging from 0 and 1, it also ranges between 0 and 1. A value of PTP of 0 corresponded to no expected thaw of permafrost. On the other hand, a PTP equal to 1 reflected a maximal difference between the two PFIs and corresponded to a high potential of permafrost thaw.



### 2.3.3 Susceptibility modelling - the DEFROST susceptibility map

The model was used to predict the occurrence of degrading permafrost over the French Alps, obtaining the so-called suscepti-bility map (e.g. Goetz et al., 2011), called here DEFROST susceptibility map. It is emphasized that the DEFROST susceptibility map does not represent the spatial footprint of degrading permafrost in its whole. Indeed, being the model calibrated on destabi-lized rock glaciers, the DEFROST map is significant only for the processes relative to destabilization of ice-rich debris slopes. Thawing of rockwalls and thermokarst formation are processes not accounted by the model. Also, the relevance of the map outside rock glaciers assumes that the processes causing rock glaciers destabilization are consistent with those causing failures in permafrost slopes. Therefore, in areas where creeping permafrost does not exist, the map may fail or be meaningless.

The DEFROST susceptibility map was computed using the R package RSAGA. Rockwalls were filtered out from the com-putation by applying a threshold slope angle higher than 35°. Also, only areas with PFI higher than 0.6, i.e. above the lower limits of probable permafrost (Marcer et al., 2017), were considered susceptible to permafrost destabilization. The model pre-dicted a DEFROST index which was classified into five susceptibility zones using the 50, 75, 90, and 95 percentiles (Rudy et al., 2017; Goetz et al., 2011). These zones described very low (<50), low (50 – 75), medium (75 – 90), high (90-95) and very high (>95) susceptibility to permafrost destabilization.

### 2.4 Recent dynamic behaviour of rock glaciers

This study aims also to get a better insight on the dynamical behaviour of the rock glaciers in the past two decades in rela-tionship to their destabilization rating. Horizontal displacement rates were estimated by manually tracking the movement of individual boulders on the surface of the rock glaciers as observed in the IGN orthoimagery. For each destabilization rating, 30 randomly selected rock glaciers were investigated by tracking one clearly identifiable boulder. The boulder was selected as the one showing larger displacements in the orthoimagery, in order to estimate the maximal displacement speed of the rock glacier. The displacement speed was then computed by dividing the distance covered by the boulder by the time elapsed between two orthoimages. Uncertainty in the estimation was quantified by evaluating the relative movements in fixed areas (e.g. bedrock, vegetalized patterns) between each orthoimages. These relative movements were due to image distortion and offset. If uncer-tainty in fixed areas was greater than detectable movements then the orthoimagery was not considered of sufficient quality and the landform was replaced by another randomly selected rock glacier.

## 3 Results

### 3.1 Destabilized rock glaciers inventory

More than 1300 surface disturbances were digitized, involving 256 rock glaciers. This indicates that more than the 50% of the active rock glaciers may be affected by some degree of destabilization (Figure 4). Of the overall population of active rock glaciers, 58 rock glaciers (11.7%) showed potential destabilization, 79 (16.1%) were suspected of destabilization and 119 (24.2%) were unlikely destabilized.



Potentially destabilized rock glaciers were mainly located in in the Vanoise National Park, which is in the Maurienne valley, and in the Queyras mountain range. In these areas, the destabilized rock glaciers were mainly found on ridges along the border with Italy.

The predominant surface disturbance we observed were cracks, which were present on 170 of the active rock glaciers. Crack clusters also had a high number of observed cases (141), while the other surface disturbances occurred in about 15% of all the examined rock glaciers. In general, the occurrence of surface disturbances were dependent on the destabilization rating (Table 3). Scarps, crevasses and erosion gullies were found in about a fourth of the potentially destabilized rock glaciers, while their occurrence decreased to about 10% on unlikely destabilized landforms. The observation of each surface disturbance was highest for potentially destabilized rock glaciers, indicating that in these landforms multiple surface disturbances coexist.

## 3.2 Recent dynamic behaviour of rock glaciers

The method used to estimate rock glacier movements was able to detect horizontal displacement rates greater than 0.3 m/yr at best, roughly corresponding to 3 – 5 pixels in the orthophotos. This limit was much higher in distorted orthophotos.

Stable rock glaciers were found to move at a maximum rate close to 2 m/yr, while potentially destabilized rock glaciers may move at up to 8 m/yr. Mean velocities ranged from  1 m/yr for stable rock glaciers to 1.5 m/yr for potentially destabilized rock glaciers for the first period (2000-2009). Although most rock glaciers did not present significant acceleration during the investigated time period, mean velocities increased in the second period (2009 – 2013) for all destabilization rating levels. About 50% of the analysed potentially destabilized rock glaciers presented strong accelerations, as some accelerated from 2 to 7 m/yr ca. Deceleration occurred in 5% of the rock glaciers and in one case a potentially destabilized rock glacier slowed down from 6.5 to 1.8 m/yr, possibly indicating the end of the destabilization phase.

## 3.3 Modelling

According to the HISTALP data, temperature rose about 1.8°C from the 1790 – 1920 to the 1980 – 2008 period. This was expected to affect a vast area of the presumed permafrost zone (Figure 6). The PTP model results suggest that the lower limit of probable permafrost would have risen by about 300 m in elevation regardless of slope orientation.

Following a stepwise backward and forward selection, the chosen DEFROST model included PISR, slope angle, elevation and curvature as predictors. In cross-validation, the mean estimated AUROC was 0.76 on the test set, indicating a good performance (Hosmer and Lemeshow , 2000). The predictors having most influence on the response variable were the PISR (AUROC change = 0.142), curvature (AUROC change = 0.070), slope angle (AUROC change = 0.039) and elevation (AUROC change = 0.016).

The model transformation functions revealed the relationships between terrain attributes and rock glacier stability (Figure 7). Surface disturbances were more likely to occur in an altitudinal range between 2700 and 2900 m a.s.l. Slope angles ranging between 20 and 40° were associated with higher predisposition to destabilization. Slightly negative to positive curvature was also favourable to destabilization. PISR was negatively correlated with the destabilization probability, indicating that rock



glacier destabilization was more likely to occur on north-facing slopes. Although not used in the final model and therefore reported for exploratory purposes only, the PTP was positively correlated with the destabilization.

### 3.4 DEFROST susceptibility map

The DEFROST susceptibility map highlights areas susceptible to destabilizing permafrost based on regional-scale model pre-
dictions (examples shown in Figure 8). The susceptibility map reproduced well the previously known cases of destabilization. The locations of the active layer detachments in Lou rock glacier were correctly represented. The collapsed areas of the Berard, Roc Noir and Pierre Brune were classified as at high susceptibility to destabilization. Nevertheless, the susceptibility map was prone to over-estimate destabilization, noticeable in areas with high index located in stable rock glaciers.

Rock glacier surfaces were investigated with respect to each susceptibility class (Table 4). About 75% of the creeping per-
mafrost was found at low or very low susceptibility to destabilization. Creeping permafrost at high and very high susceptibility to destabilization accounted 10% of the total creeping permafrost surface, i.e. 2.8 km$^2$. While about one third of this surface was located in potentially destabilized rock glaciers, more than 1.4 km$^2$ of stable and unlikely destabilized rock glaciers were found at high and very high destabilization susceptibility.

## 4 Discussion

### 4.1 Rating rock glacier destabilization

The present study provided the first comprehensive assessment of rock glacier destabilization for the French Alps, suggesting the high prevalence of the phenomena in this area. Destabilized rock glaciers were more likely located in the Maurienne Valley, Vanoise National Park and Queyras range. In these areas the densely jointed lithology was suspected to generate mainly pebbly rock glaciers (Matsouka and Ikeda , 2001). This suggested that destabilization may be more likely to develop in pebbly rock
glaciers, as observed in the Berard and Lou rock glaciers. However, recognizing surface disturbances on pebbly rock glaciers may be easier than in "blocky" rock glaciers, as smaller cracks are more evident. This may create a bias which should be studied more in detail by investigating geomorphological features of destabilization occurring on blocky rock glaciers.

Rock glacier destabilization rating can be a relevant tool for the local authorities to assess risks related to the degradation of the periglacial zone, as we identified all rock glaciers presenting signs of destabilization in the region. The destabilization
rating suggests the severity of the potential hazard and can help identify actions that should be undertaken to deal with the problem. In general rock glaciers with low destabilization rating are currently evolving slowly or are stable, and consequently monitoring based on remote sensing may be sufficient. Suspected or potentially destabilized rock glaciers require more caution and in-situ monitoring is recommended.





### 4.1.1 Uncertainties in rating rock glacier destabilization

A potential source of uncertainty in this study was the subjectivity that can occur while mapping surface disturbances and rating the degree of destabilization. These activities were based on expert knowledge; however, it is possible that mapping and rating results vary depending on the operator. For example, the operators in charge of the digitization process were requested
to interpret surface features that in many cases have small dimensions with respect to the resolution of the orthoimages, making the identification challenging. Also, although surface disturbances were inventoried into the catalogue in an attempt to standardize the classification, destabilized rock glacier morphology is complex, and its identification requires intense training. In many cases the boundaries between the different typologies proposed were not sharp. Personal knowledge of the process evolved through the inventory compilation, requiring various iterations to review the work. Another issue was that the operator's
metrics of judgement varied through the process, as the classification might get stricter (or looser) when the operator deals with a series of destabilized (or stable) rock glaciers. The ratings were compiled and revised by different operators in an attempt to mitigate these effects. Some cases were subject of debate, highlighting significant individual biases. These biases can influence the resulting susceptibility model (Steger et al., 2016). It is therefore strongly recommended to integrate the inventory with in situ observations when possible and to maintain a critical attitude towards the data. At present time France does not have a
LiDAR-based high-resolution DEM covering the study region. Such data could be used to revise in the inventory in the future in order to reduce errors due to poor quality of the orthophotos.

    Although observing aerial orthoimagery or high resolution DEMs could not replace the relevance of a proper in-situ survey, it provides us with data and resulting insights that would normally not be possible with in-situ surveys alone, a characteristic that fitted with the aim of the study. Additionally, the use of orthoimagey has been proved to be a useful approach for mapping
rock glacier surface disturbances by Serrano (2017), where the results of field observation were compared to observations from orthoimagery. Although Serrano (2017) investigated a limited number of sites, those results were encouraging, showing that the method was relevant. The use of multiple orthoimages was believed to successfully reduce subjectivity-related issues in most of the cases. Observing the movements of the landforms was a valuable decision support tool, as surface disturbances could be related or not to discontinuities in a pronounced displacement field. Also, the use of multiples orthoimages reduced
potential errors due to bad lighting that may enhance features that may be unrelated to destabilization processes (Serrano , 2017).

### 4.2 Recent rock glacier dynamics

The range of rock glacier velocities was in agreement with previous findings. A rock glacier normally moves with a rate of 0.1 to 1-2 m/yr (Roer et al., 2005), reaching up to 5-10 m/yr in extreme cases of destabilization (Delaloye et al., 2008). The ability
of velocity to discriminate rock glacier destabilization was measured by using multiclass AUROC (Hand and Till , 2001). Results indicated that velocity is a good predictor (AUROC : 0.72) to discriminate the rate of rock glacier destabilization. However, it was found that potentially destabilized rock glaciers may show relatively normal velocities, down to 1 m/yr. This can be a significant finding since velocity is usually used as the main criterion for spotting and monitoring potentially





hazardous rock glaciers (e.g. Barboux et al., 2013). Although we do not question the relevance of fast-moving rock glaciers in identifying potentially hazardous rock glaciers, in our study we observed that potentially destabilized rock glaciers seemed to be more prone to strong acceleration than stable rock glaciers, as about half of these landforms doubled their speed within the past two decades. This may indicate that destabilized rock glaciers may have an unexpected dynamic behaviour in the

short term (Delaloye et al., 2008). Thus, the authors suggest that also "slower" rock glaciers that present evidence of ongoing destabilization may also be potentially hazardous.

### 4.3   Modelling destabilizing permafrost

Despite the various limitations of the database, results were encouraging. The spatially cross-validated model had a good performance. The relationships with predictor variables were found to be consistent with topographic settings observed in

known cases of destabilization. Slope angle and convexity relationships were coherent with field observation, suggesting that steep slopes and flat to convex topography are suitable to the development of surface disturbances. The PTP was positively correlated with the DEFROST index, indicating that destabilization was more likely to occur where the permafrost belt was expected to be thawing.

PISR had the most importance in the model, suggesting that rock glacier destabilization is primarily more likely to occur

on north facing slopes. In an investigation of active layers detachments, Rudy et al. (2017) obtained the similar result for permafrost in the Canadian arctic. They suggested that the greater occurrence of active layer detachments on north facing slopes may be due to how the longer lasting snow cover on northern slopes may enhance soil saturation, which an important trigger for the active layer detachments. This explanation may also be valuable in the context of the present study, as water infiltration is also a relevant factor causing destabilization of rock glaciers (Ikeda et al., 2008).

### 4.4   The DEFROST susceptibility map

Overall, permafrost destabilization was adequately described, as indicated by the cross-validated performance, in most of the observed cases of destabilization. Although cases of potential destabilization were inventoried, rock glaciers that have a low rating of destabilization and are located in areas with high DEFROST susceptibility should be identified as having a high potential of showing future destabilization. Results indicated that these rock glaciers had a large area of high susceptibility

to destabilization and should be monitored for risk assessment. In particular, the Laurichard rock glacier is a site currently under monitoring which was found to present a medium to high susceptibility to destabilization in this study (Bodin et al., 2008). The comparison of the future evolution of this landform with respect to the DEFROST susceptibility map is therefore recommended.

### 4.5   Assessing the spatial footprint of degrading permafrost

In this last section we propose a quantification of the total surface of degrading and destabilizing permafrost in the region. This was done by extrapolating the PTP and DEFROST indexes in non-rock glacier surfaces under the assumption that processes at




the core of permafrost existence and degradation in rock glaciers hold in non-creeping permafrost. The authors acknowledge that rock glaciers have a special thermal regime and peculiar dynamic characteristics due to their structure and ice content. Nevertheless, mechanisms causing rock glacier destabilization, such as active layer thickening, loss of cohesion and resistance to water erosion, were believed to be playing a role in debris flow initiation and scree slope failures (e.g. Haeberli et al., 1997).

Further work to assess the validity of the DEFROST susceptibility map in non-creeping permafrost is strongly encouraged. This can be done using a different approach, e.g. by analyzing debris flow occurrence in periglacial watersheds, as proposed by Damm and Felder (2013).

This quantification highlights widespread permafrost degradation and destabilization susceptibility in the region. Discontinuous permafrost covers about 770 km$^2$ of the French Alps (Marcer et al., 2017). Almost 50% of this surface was predicted to

be unsustainable in the present climate (i.e. PTP > 0.8). The DEFFROST map extrapolated to non-rock glacier areas indicated that conditions highly and very highly susceptible to destabilization can be found in over 60 km$^2$, involving about 8% of the permafrost zone.

## 5   Conclusions

The present study aimed to give insights into the extent of degrading permafrost in the French Alps. This was done by mapping

and modelling rock glacier destabilization in the region using orthoimagery collection, 25 m x 25 m resolution DEM and statistical modelling. This methodology carried several limitations, due to subjectivity and modelling issues. Therefore, absolute model performances and the appearance of the susceptibility map may not be exact and further work is strongly encouraged. Integrating the observations with high resolution LiDAR DEM and with new field-observations could spot possible systematic biases in the destabilization rating attribution and significantly reduce uncertainty.

Despite the limitations of this methodology, the study contributes to the knowledge of periglacial risk in the French Alps. The destabilization of creeping permafrost was found to be a widespread phenomenon which involves more than 10% of the total surface of active rock glaciers, i.e. 3 km$^2$ ca. Only half of this surface was attributed to rock glaciers currently showing a relevant degree of destabilization, suggesting that several stable rock glaciers are good candidate to experience destabilization in the future. Furthermore, permafrost degradation and destabilization may affect 50% and 8% of the permafrost zone respectively.

These findings suggest that mountain permafrost in the region is in a critical state, possibly enhancing periglacial risks. In this context, the present study contributes by having mapped potentially destabilized rock glaciers and areas considered susceptible to destabilization. In this sense, we suggest that the modelling framework proposed is relevant and further efforts to better acknowledge the phenomena are strongly encouraged.

*Code and data availability.*   The R code to model rock glacier stability and database is available and built in RGUI version 3.4.4. Shape files

for surface disturbances (one file per feature type. Data are in .shp format) and PTP and DEFROST susceptibility maps are available (.tiff format). Data are in referenced in EPSG : 2154.





*Competing interests.* Herby we declare that no competing interests is present for this study

*Acknowledgements.* The present study was funded by the region Auvergne-Rhone Alpes through the ARC-3 grant and by the European Regional Development Fund (POIA PA0004100) grant. The Lanslebourg - Val Cenis municipality also contributed to the present study by funding internships within the PERMARISK project.



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



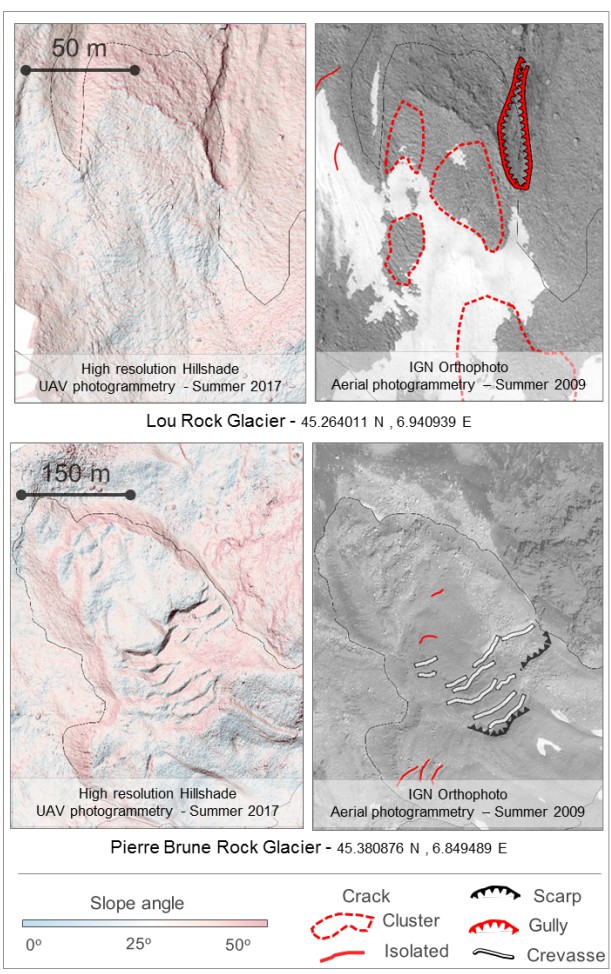

**Figure 1.** High resolution hillshades and othophotos acquired by UAV imagery (DJI Mavic Pro) of two rock glaciers used to calibrate the catalogue of destabilization evidences. On the Lou rock glacier could be observed crack clusters and a debris flow gully. On the Pierre Brune rock glacier could be observed several deep crevasses and scarps.



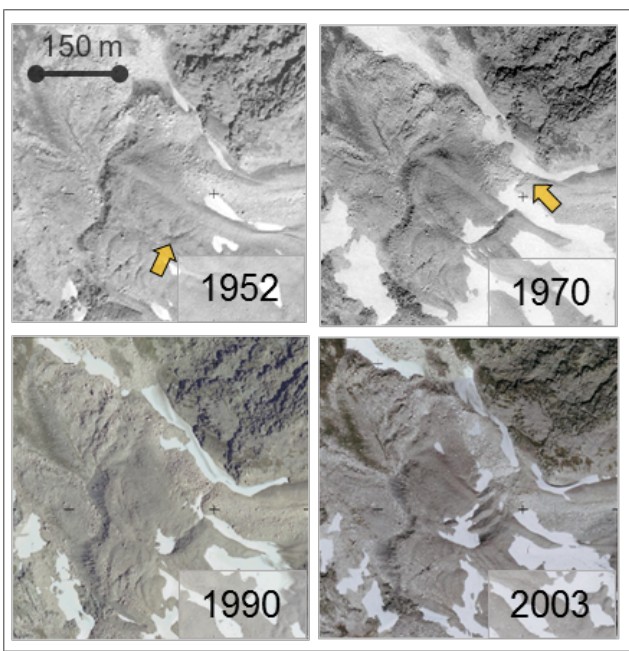

**Figure 2.** The evolution of the destabilization of the Pierre Brune rock glacier. The destabilization evidence, in this case a crack observable since 1952, evolved to a crevasse, observable in 1970. Afterwards, the landform was stable for 20 years as destabilization evidences did not further evolve. Between 1990 and 2003 the rock glacier experienced severe destabilization with the formation of new crevasses and scarp in the location of the 1952 crack.





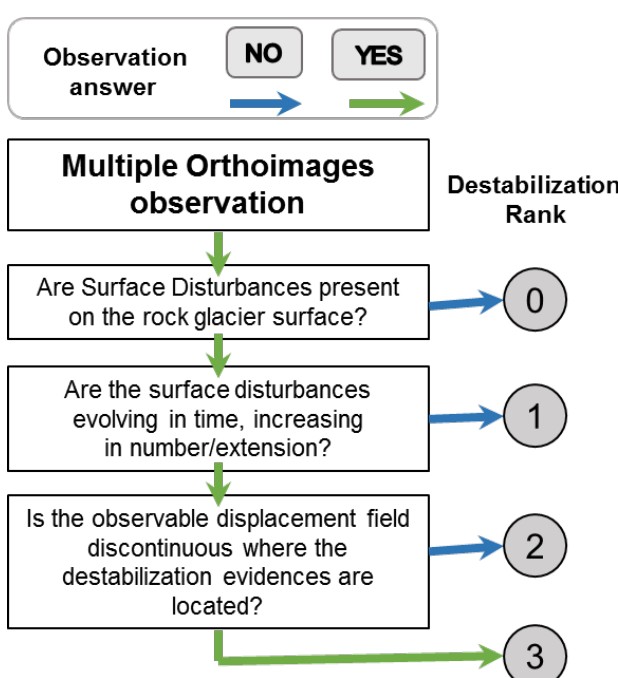

**Figure 3.** General pipeline used to rate rock glacier destabilization by observing surface disturbances and qualitative displacement field. Higher rates of destabilization indicate potentially unstable rock glaciers, while lower ratings indicate stable rock glaciers.



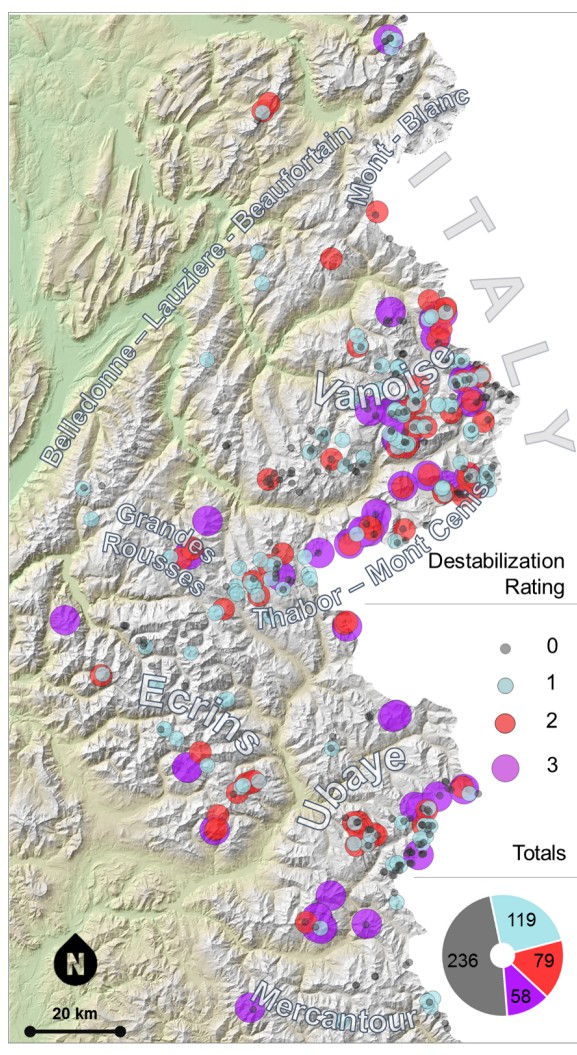

**Figure 4.** Map and pie chart of destabilized rock glaciers in France classified by rock glacier destabilization rating. On the map are reported the major mountain ranges of the region. Potential destabilization is widespread in Vanoise, Thabor-Mont Cenis and Ubaye mountain ranges.





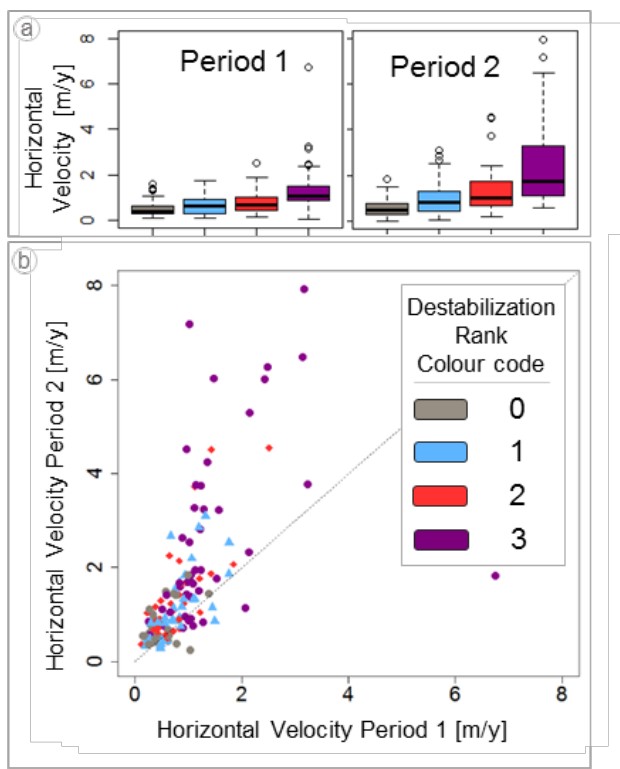

**Figure 5.** Summary of the rock glacier velocities in the past two decades according to their destabilization rating. Velocity refer to fastest boulder traceable on the orthomosaic collection. On top (a), boxplots of the observable boulder velocity per rock glacier according to their destabilization rating in in the period 1 (2000 – 2004 to 2008 – 2009) and period 2 (2008 -2009 to 2012 -2013). On the bottom (b), velocity comparison between the two periods. Points above the dashed line indicate accelerating rock glaciers.





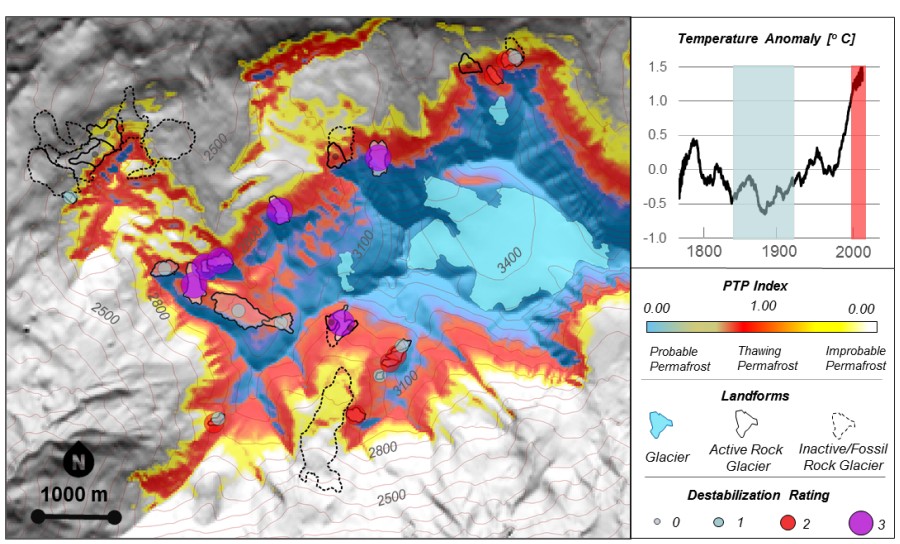

**Figure 6.** Example of Potential Thawing Permafrost (PTP) distribution in the Roc Noir sector, indicating the extent of the permafrost zone not in equilibrium with the present climate (red colored areas). Temperature warming to compute the map is evaluated using HISTALP data (Auer et al., 2008) between the end of the Little Ice Age (light blue shade period in the temperature anomaly plot) and the current climate (red shade period).





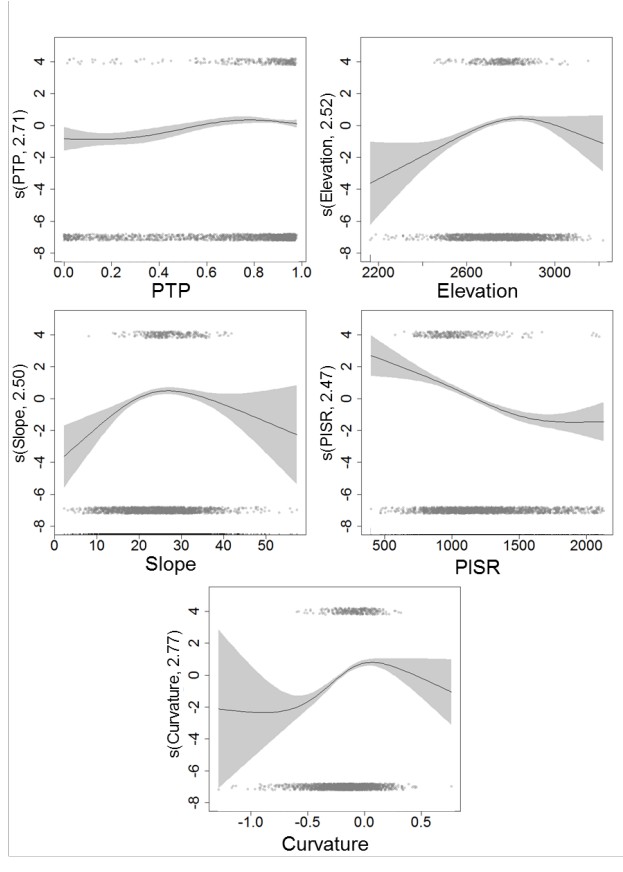

**Figure 7.** Transformation function plots of the GAM model showing the relationship between each predictor variable and destabilization occurrence. Data distribution with respect to predictor variables are indicated with dots on top (destabilization evidence) and on the bottom (stability evidence) of the plots.



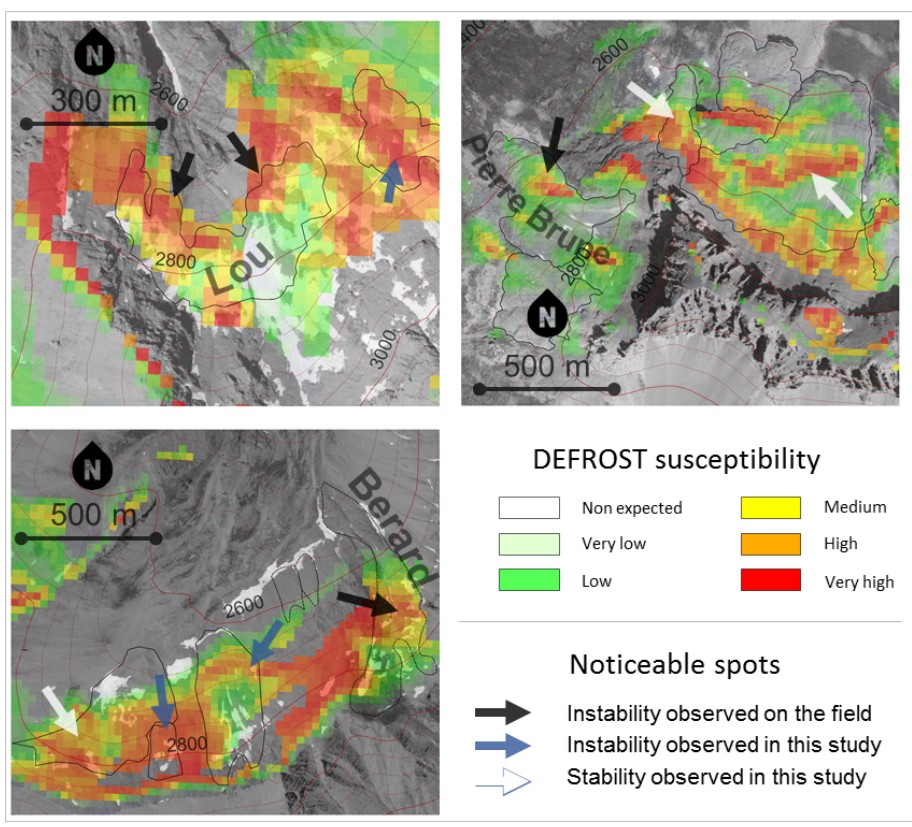

**Figure 8.** Examples of DEFROST susceptibility map in Lou, Pierre Brune and Berard and neighboring rock glaciers. The DEFROST susceptibility map successfully identifies instabilities observed either on the field and by observing the orthomosaic collection (black and blue arrows respectively). Nevertheless, some predicted instabilities were observed in areas that appear stable by observing the orthomosaics (white arrows).



**Table 1.** Description of surface disturbance features that could be observed in the field or from orthoimagery to identify signs of rock glacier destabilization

| Feature | Description |
| --- | --- |
| Debris flow gully | These are erosion features that mark the frontal lobes of the rock glacier (Delaloye et al., 2008). Although frontal erosion in some topographic context is a natural process, in case of destabilized rock glaciers, it reveals abnormal amounts of loose material that are expelled as debris flow of significant intensity (Kummert and Delaloye , 2018; Schoeneich et al., 2017). In the Lou rock glacier, the debris flow gully caused by an active layer detachment is more than 2 m deep and 10 m wide. The gully is identifiable on orthoimages thanks to its darker matrix below the initiation point. Lateral and downstream deposits characteristic of debris flows are also often identifiable. |
| Cracks | These are shallow linear incisions in the surface of an active rock glacier where a strain is applied (called "scars" in Roer et al. (2008)). Cracks can be several tens of meters long and occur either individually or in a great number, being spaced from each other of only few meters. In this case we define the feature as a "crack cluster", i.e. the "rugged topography" proposed by Roer et al. (2008). Their proximity and shallowness lead to the assumption that they affect only the active layer of the landform. Nevertheless, this feature was found to be largely predominant on the Lou (Schoeneich et al., 2017) and Tsate'-Mory (Roer et al., 2008; Lambiel, 2011) rock glaciers and therefore considered of interest in the context of the study. |
| Crevasses | These deep transverse incisions on the rock glacier surface can range in length from several meters to the entire landform width (Avian et al., 2005; Delaloye et al., 2008; Roer et al., 2008). Their depth is substantially larger than the active layer thickness, suggesting the presence of a shear plane sectioning the frozen body. Crevasses may be isolated or grouped. Spectacular crevasses can be found on Pierre Brune rock glacier (Fig. 1), where they are up to 7 m deep and 10 m wide, cutting across the entire landform (about 150 m). Similar dimensions are reported in the Furggwanghorn rock glacier (Roer et al., 2008). |
| Scarps | Described by Scotti et al. (2016) and Delaloye et al. (2008) as steep slopes (30 to 40°) several meters high cutting transversally the entire rock glacier. Scarps are associated with deep shear planes that disconnect the rock glacier into two bodies that creep at different speeds. Their activation is associated with a sudden acceleration of the downstream portion of the landform. One of the biggest scarp observable in the region is the one on Roc Noir rock glacier (Serrano , 2017). This S-shaped scarp, 20–30 m high and 40–45° steep, cuts transversally the whole landform (120 m) and the downstream lobe creeps about twice as fast as the upper part. |





**Table 2.** Rating classes used to describe rock glacier destabilization

| Rating | Label | Description |
|---|---|---|
| 3 | Potential destabilization, potentially destabilized rock glaciers | Surface disturbances are well recognizable and evolve in time, increasing in number and/or size. The deformation pattern of the rock glacier is discontinuous and some sectors move significantly faster than others. The source of the discontinuity may be located at the rock glacier's root and the whole landform may be affected by destabilization. Dynamical discontinuities are sharp and coincide with the presence of surface disturbances. Sectors moving appreciably faster may also present a series of surface disturbances. |
| 2 | Suspected destabilization | In these landforms the surface disturbances are well recognizable and evolve in time, by increasing in number and/or size. The velocity field is continuous, i.e. there are no abrupt spatial differences in the velocity field. If there are sectors moving faster than others, their transition is smooth |
| 1 | Unlikely destabilization | In these landforms surface disturbances do not appear to evolve in time. The rock glacier presents a continuous dynamical field, with no sectors moving substantially faster than others. |
| 0 | Non-observable destabilization | Active rock glaciers not presenting surface disturbances are considered as stable. |





**Table 3.** Number of rock glaciers per destabilization rating showing a specific surface disturbance.

| Destabilization rating | Number of Rock glaciers | Scarps | Crack Cluster | Crack | Crevasses | Gully |
|---|---|---|---|---|---|---|
| Unlikely | 119 | 8 | 52 | 86 | 13 | 11 |
| Suspected | 79 | 10 | 44 | 43 | 12 | 11 |
| Potentially | 58 | 17 | 45 | 41 | 13 | 15 |
| Totals | 256 | 35 | 141 | 170 | 38 | 37 |



**Table 4.** Active rock glacier area per class of DEFROST susceptibility.

| Destabilization Rating | Surface per DEFROST susceptibility class [km$^2$] | | | | |
|:---:|:---:|:---:|:---:|:---:|:---:|
| | *Very Low* | *Low* | *Medium* | *High* | *Very High* |
| **0** | 6.98 | 2.98 | 1.60 | 0.43 | 0.32 |
| **1** | 3.79 | 2.07 | 1.34 | 0.38 | 0.35 |
| **2** | 2.22 | 1.36 | 0.73 | 0.27 | 0.29 |
| **3** | 0.40 | 0.57 | 0.61 | 0.36 | 0.48 |
| **Cumulative Surface** | 13.39 | 6.97 | 4.28 | 1.44 | 1.44 |