# Peer review of "Evaluating the destabilization susceptibility of active rock glaciers in the French Alps"

_The Cryosphere, 2018_

## Referee Comment (RC1) · Anonymous Referee #1 · 11 Jul 2018

General Comments

This paper uses a generalized additive model to model rock glacier stability resulting in a susceptibility map highlighting potential areas of permafrost destabilization. This is possible as the stability of rock glaciers is a reflection of the underlying permafrost stability. Results from this research are an important contribution to improving our understanding of periglacial risk and of broad significance in mountain permafrost regions. However I do have a number of comments that should be addressed before moving forward with publication.

[Figure]

The introduction could be improved. The relationship between rock glacier stability and permafrost destabilization needs to be more concisely and clearly presented.

Some key information about the study area is missing such as general information on ice content and active layer characteristics. There are a number of sentences that allude to its importance as observed from other papers but there is nothing specific to this paper. If this data does not exist this should be acknowledged.

It may be useful to have a Data section in the Methods where you introduce the different orthoimagery datasets and their time periods, resolution and source, and, the DEM, including its resolution, source etc. I found myself having to jump around to look for this information.

The methods explaining the models can be improved. The introduction of the GAM is ok, but it is missing some critical information. For the GAM, what type of smoothers were used and did you control for their flexibility, what were the degrees of freedom? This information is important particularly for overfitting which depends on the flexibility of a smoother, which can be controlled by the degrees of freedom. In the methods you state you will examine the accuracy of the model using sensitivity and specificity, this is not followed up on in the results.

Some more work needs to be done to better lay out the relevance of the selected terrain variables and their connection to rock glaciers/permafrost degradation. You try to link permafrost degradation to rock glacier activity and while you do provide evidence of this through the literature the link within your own study is not clear. I assume you are using the terrain variables (i.e. PISR, slope etc) as possible proxies for surface/subsurface conditions i.e. ice content, active layer thickening. . . etc.? Often terrain variables are selected as they serve as proxies for surface conditions that are difficult to represent through spatial datasets, however this connection is not well made throughout the methods, results or discussion.

The way you present the methods used to select the points to build the model is unclear. You say 5 randomly selected points (the size is not stated) were used to extract terrain information from each rock glacier, and that this was done as there were only 58 potentially unstable rock glaciers compared to 79 suspected destabilized rock glaciers and 119 unlikely destabilized rock glaciers. You also state that the response variable, which is representative of stable and unstable rock glacier zones were then assigned a defrost index of 1 or 0. Where those 5 points, which I assume were classified as potentially unstable, collected to increase the unstable (1's) inventory? If yes, and if you then extracted the terrain information for each of those 5 points, it's possible that the model is biased towards terrain information specific to the potentially unstable rock glaciers represented in your study. This could reducing the overall usefulness of the model when applied to the rest of landscape or perhaps lead to an overestimation of areas modeled as potentially unstable.

The addition of a table outlining the size of the inventory (1's and 0's) that was used to build the model would be very helpful. This may make it easier to present the different methods i.e 5 points being extracted from unstable rock glaciers. Currently, it is not easy to get a handle on the size of the inventory and how this is integrated into the model.

Specific Comments

P1 L3 The description of the imagery used in the abstract does not match with what is presented in the methods. In the following sections you state you have imagery from 2012-2013.

P1 L6 Be more specific on the time periods being used to compare the deformation patterns.

P1 L11 On P9 L31 you state that the slopes associated with higher destabilization rates is 20-40 however here in the abstract you only state that up to 30. Which is correct?

P2 L16 What methodology? No methodology is mentioned but I assume you are referring to methodology in the Sattler et al. paper? Suggest stopping sentence at "...degrading permafrost" or briefly layout methods of paper that show that the initiation points of debris flows weakly correlate with the spatial footprint of degrading permafrost.

P4 L5 Additional details on the multi-temporal orthoimagery should be included i.e. dates, resolutions, source (satellite/airplane/UAV). Is this the same imagery that is introduced in section 2.2, if yes, some effort should be made to rework/better connect this information.

P4 L5 I see in the discussion you address errors associated with your mapping however, you should include a sentence in the methods explaining how you plan on assessing this.

P4 L5 What features are you using to attribute a classification of active or inactive?

P4 L6 What was the final inventory of active rock glaciers? This should be included here.

P4 L13 Can you be more specific or provide better descriptions of the different surface disturbances, their morphology and triggering causes. As it currently reads, it is very vague.

P4 L 222 In the Figure 1 caption you say you used UAV images to map distinct destabilization features. But when I read the description here, there is no mention of UAV imagery. While the imagery you state you have is fairly high resolution, mapping cracks with 2 m imagery vs. mapping cracks with 50 cm imagery can be quite different. Were there resolution issues when mapping? How confident are you that you mapping all the features equally? Do you have any metrics on your mapping error? I see that you acknowledge this in your Discussion but like I mentioned in an earlier comment, a couple of sentences should be included in your methods addressing how you plan to do this.

P4 L25 What is the biggest factor, image quality or the availability of multi-temporal images? What is the minimum resolution needed to map these features, are your images high enough resolution?

P5 L5 Oblique photography? This is the first time this is mentioned, is this different than the imagery presented earlier?

P5 L7 You finished the previous paragraph stating that you will present a slightly different definition of a destabilized rock glacier. I think the start of this paragraph should begin with a clear and concise explanation of your definition followed by the examples/observations you use to support it. As it currently reads I do not actually know what your definition a destabilized rock glacier is. A definition needs to be presented before you describe your destabilization rating on L24.

P5 L24 Be more specific, what combination of surface disturbances/qualitative assessment of recent deformation patterns merits a rating of 1 or 2...?

P5 L29 What was specifically modelled in arctic permafrost, this description is very vague.

P6 L5 If these 5 points were extracted from a potentially unstable rock glacier (1-presence), I assume all 5 points were represented as 1's in the model?

P6 L5 What size were the multiple points that were extracted from the rock glaciers. Are you confident there is no overlap of these points? You say that model performances stabilized for more than 5 points, how was this discovered? What was the minimum and maximum number of points used? Was model performance assessed for the different number of points using only the best model?

P6 L11 Was the forward and backward stepwise variable selection not used to select the best multiple variable model (model with the lowest AIC)? The way it currently reads is that you populated a number of models with different combinations of variables, found the one with the lowest AIC and then used stepwise (forward, backward) variable

selection to identify which predictors were the best in the what would already be the best model. Suggest rewrite for clarity.

P6 L18 What was the size of each cluster, how many rock glaciers in each?

P7 L5 Assigning your response variables a DEFROST index and then assigning a DEFROST index to your DEFROST susceptibility map decreases the clarity of inputs and outputs for the model. I would refer to your response variables as either values of 0 for stable or 1 for potentially destabilized rock glaciers

P7 L8 What was your final count of 1's and 0's used to train the model? This needs to be included in the text

P7 L11 Probability of thawing permafrost?

P7 L15 The DEM is a bit on the course side, do you think this could have affected your results? What is the average size of the rock glaciers? This general information of size, stability, mapped destabilization features should be included as a table

P7 L20 Would there be a complete absence of snow cover throughout the summer? For what periods of the year was PISR computed?

P7 L21 – 34 I don't think the description of PFI should go in the Model predictor variable section. It is not actually used as an input to the model but is almost used a threshold for which you limit your resulting susceptibility map to.

P7 L21 It doesn't appear that you evaluate potential permafrost thaw using analytical methods presented by the others (cited in text) in this paper. You state that "Here, we used the Permafrost Favourability Index. . ..). I would remove the first part of the paragraph as it doesn't appear useful in the context of this paper. Also was the PFI recomputed specifically for this paper or are you using the results from your 2017 paper. If you are, you do not need all of the details that you include in this section.

P7 L29 What do you mean when you say the "The resulting map, which corresponded

Interactive
comment

to a theoretical permafrost distribution in equilibrium with the current climate, was finally subtracted from the PFI, obtaining the Potential Thawing Permafrost zone"? In the previous sentence you say the PFI map was recomputed using the model parameters and then in this sentence you say the resulting map was subtracted from the PFI...? If you are subtracting two PFI maps, how are these maps different?

P8 L28 Suggest you move the sentence, "This indicates that more than 50%...." to after the next sentence where you actually provide the percentages of unstable glaciers.

P9 L10 This is just personal preference but it makes the paper easier to follow if the results are presented in the same order as they are presented in the methods. Methods related to section 3.2 were the final section of the methods but here are presented as the second set of results.

P9 L10 How many glaciers was this analyse done on?

P9 L11 If the orthoimages have resolutions ranging from 0.5 to 2 m how are displacement rates of 0.3 m/year detected and how does this correspond to ∼3-5 pixels? Is this possible because dates have been grouped and zero movement is inferred in the missing years. If not, how are you accounting for movement in the years you don't have imagery?

P9 L12 What was the limit in distorted orthophotos?

P9 L15 It appears as though you have grouped the first two periods, 2000 – 2004 and 2008 – 2009. This should be stated in the text. Again, how are you inferring movement for the years you do not have data? It seems like a bit of a reach to present the second period as 2009 – 2013 when you only have data from 2012 – 2013. That's a big chunk of time with no data.

P9 L21 Unsure as to why the PTP model results are with the modelling results?

P9 L25 Earlier you mention that you also used sensitivity and specificity to assess the model performance, where are these results?

P10 L8 Why do you think the model is overestimated these areas? This should be expanded on in the Discussion section on the susceptibility model.

P10 L18 Do you think that adding a surficial geology variable or a variable that highlights jointed bedrock would be useful?

P11 L 2 – 16 This is great and addresses a number of my prior questions. It is great that you acknowledge the challenges but are you able to quantitatively provide an idea of the error? Was any field validation done for any of the mapping?

P11 L30 Multiclass AUROC, what do you mean here. Velocity isn't used in the model so if you evaluated other variables in a different way that should be presented in the methods and the results.

P12 L8 This sections needs to be strengthened. What else to these variable tell us about process? I already made this comment in the General Comments section but I will state again that more work needs to go into explaining the importance or the terrain variables or the potential surface processes they represent.

P12 L12 Was PTP strongly correlated with the DEFROST index? At what susceptibility class was it most correlated with i.e. are areas modelled as high susceptibility the areas where the permafrost belt is expected to be thawing most?

P12 L15 Is it possible that north-facing slopes may have greater ice contents closer to the surface in part due to increased soil saturation but possibly also due to shallower active layer depths? In your area do you know how ice content varies or if there is a relationship with aspect? Also, did you look at the terrain information for your glaciers? Are there more glaciers perhaps on north-facing slopes? This would be worth looking at and presenting.

Technical Corrections

P1 L12 Model performance should be singular

P2 L13 Do you mean altitudinal?

P2 L9 change relationship to relation

P2 L15 Suggest remove permafrost from permafrost initiation points. . .

P2 L23 Remove "so-called"

P2 L24 I would suggest changing dynamical to dynamics throughout. I recognize that this term may be more common in your field but it doesn't fit in every place it is used.

P3 L19 1500 m a.s.l

P4 L3 Improve the sentence. . . "Although activity. . ."

P4 L4 Suggest changing the word "noticed"

P5 L23 patches

P5 L31 Remove (response variable) is it redundant.

P7 L30 Suggest changing The so-called "melting area". . . to the "thawing area"

P8 L2 remove so-called, what else would the map be called?

P8 L28 . . .involving 256 active rock glaciers

P8 L23 vegetation patterns

Figures

A study area map is needed. You refer to a number of specific locations ex. P9 L1 Vanoise National Park. The study area map should include these areas to put this research into context. You could possibly use Fig 4 but this figure needs to be put into a larger geographical context as well.

Figure 1 – The hillshades and orthophotos were both acquired by UAV imagery? This is the type of imagery that is available for the different temporal periods for all of France?

Figure 2 – Add arrows to the 1990 and 2003 images to point out changes as well

Figure 3 – Clean up spelling. . . Surface Disturbances doesn't need to be capitalized

Figure 4 – Refer to this more in the text

Figure 5 – It doesn't seem as though the boxplots and the scatter plot are both needed. The scatterplot does a good job portraying the differences in velocity between the periods

Figure 6 – The hillshade on the DEM makes the landscape look inverted, higher elevation areas appear as valleys. This should be corrected. The circles representing the destabilization rating are difficult to pick out for a lot of the rock glaciers, particularly those with a lower rating, I would suggest making the outline of each point thicker and perhaps a darker colour.

Tables

Table 3 – Beside each descriptor put the numerical destabilization rating

―――――――――――――――――

---

## Referee Comment (RC2) · Anonymous Referee #2 · 18 Aug 2018

General comments:

Good paper on an subject gaining in importance in the understanding of the behavior (stability) of mountain debris slopes underlain by permafrost : the so-called destabilization of rock glaciers. On the basis of an extended original dataset the authors spatially model the susceptibility of a location to be affected by a destabilization process in the French Alps. The preparatory work of inventorying the destabilization indices on rock glaciers in the entire French Alps is impressive and constitutes for me the most attractive part of the paper. The statistical modelling approach appears to be good, but can

only provides results which are very difficult to validate in my point of view. It is so partly less convincing. Some of the terminology used in the paper (permafrost destabilization, rock glacier destabilization, stable/unstable rock glacier, hazardous rock glacier) is somewhat unclear and even questionable. It has to be checked carefully all the paper along. I also have some questions about the interpretation/use of some the destabilization indices. Maybe some of the results may change in accordance. After having been revised the paper will be definitely very worth of being published in TC.

I hope my comments/suggestions can be useful for the authors.

Specific comments:

Title is not good. "Inferring the destabilization susceptibility of mountain permafrost. . ." has no real sense, permafrost being a thermal phenomenon. This is not the scope of the paper, which is conversely dealing with the mechanical "destabilization" of rock glaciers. There is however a much better alternative proposed by the authors on P3 L15, which can give a title like "Evaluating the destabilization susceptibility of active rock glaciers in the French Alps".

Abstract : may have to be adapted after revision of the paper

P1 L17 Express what is meant precisely by (widespread) permafrost degradation. This is not clear at all, but a very important concept for this paper. Check then in the whole paper if the concept is used always exactly with the same sense. Prefer "Permafrost has shown signs of w. degradation for the past decades in the European Alps".

P1 L18 : The connection between air temperature and ground temperature is tricky (snow buffering effect) and I do not really understand the meaning of "extreme warm air temperature" in the sentence. I would suggest to simplify it as "Warmer climatic conditions are expected to cause. . ." (eventually is a redundancy and can be omitted). Is there not more recent and more adapted references ? Finally, I do not see the link with the previous sentence.

P1 L20 : The "thermal inertia" is in particular highly related to the ice content, which can be relatively high in a rock glacier. This should be mentioned. Ground instead of soil. P1 L21 : I do not understand the meaning of this part of the sentence in the context of the present study. Why currently ? Is really the reference adequate ? Would not be for instance Scherler et al. 2013 (https://doi.org/10.1002/jgrf.20069, see in particular fig. 5, where the modelled impact of climate warming on two sites with very contrasted ice content is illustrated) more appropriate ?

P2 L1 : What kind of "other processes" ?

P2 L3 : Rock glacier destabilization can be caused by other factors than only climate-induced permafrost warming (e.g. cited Roer et al. 2008, Delaloye et al. 2013). The sentence must be adapted in consequence. This is a very important point, because the susceptibility model appears to be based on the assumption of a climate impact only.

P2 L4 : Delaloye et al. 2013 (and not 2008). To be changed also further in the paper.

P2 L4-5 : These events are far from all "representing a serious threat for alpine communities". Sentence to be adapted.

P2 L5 (and many times in the paper): Permafrost destabilization. What is this ? The authors are rather talking about the destabilization of frozen ground inducing almost significant mass movements (>100'000m3 ?). Permafrost destabilization appears to be an inadequate terminology that must be replaced by rock glacier or debris slope destabilization and adapted all the paper along.

P2 L7, L9-13: Permafrost degradation. Again, what are we talking about ? About a complete ice melt = permafrost has disappeared, the temperature is now above freezing point ? Or about an increased liquid water content (partial ice melt) by warmer permafrost temperature (without any permafrost thaw) ? Is a permafrost warming from -1 to -0.5°C consecutively increasing the liquid water content a permafrost degrada-
tion ? In my point of view yes, making that almost all permafrost in the Alps (and in many places elsewhere) is currently degrading ! L13 : It looks that permafrost degradation is here considered as where permafrost is still occurring where it should not be (that is, despite current ground surface thermal conditions that could no more permit its occurrence) ?

P2 L14 : What are stable and unstable rock glaciers ? In addition, I cannot agree with the sentence, which seems to be based on the assumptions that all rock glacier destabilizations are induced by climate warming (permafrost warming) and that all rock glaciers with "degrading" permafrost conditions have to destabilize.

P2 L20: "Observing rock glacier dynamics and morphology can be rather useful" for what ?

P2 L21: Permafrost degradation (complete ice melt ?) in ice-rich landforms does not directly cause the mobilization of significant amount of materials. It makes the material easily erodible, but does not put it in motion. It does not trigger debris flow, but only precondition it. Moreover, this is the (increased) motion of the rock glacier, which is making (more) materials available for later debris flow events (e.g. Kummert)

P2 L23 : An increase of the liquid water content is assumed to cause the so-called destabilization (and not can).

P2 L24-29: About the occurrence of destabilization of active rock glaciers, see also Lambiel et al. 2008. Proceedings of the Ninth International Conference on Permafrost, Fairbanks, Alaska, 1 pp. 1019-1025, in particular Table 2 and related text.

P2 L26 : . . . exceptionally (instead of eventually ?) lead to the collapse of the rock glacier (or a significant part of it)

P2 L29: Lambiel and Reynard (2001) has nothing to do with destabilization

P3 L13: Is DEFROST the most appropriate name for the model, because it helps to evaluate the destabilization susceptibility of active rock glaciers only, and not permafrost (or all permafrost slopes) ?

P3 L19 : Sorry but 15'000 km2 fits with the total area of the French Alps (50-75 x 250 km) and consequently not with the area above 1500 m. And why to mention this latter area ?

P3 L22: Climate is changing fast. Indicate the reference time period for the elevation values of the annual $0°$ isotherm.

P3 L23: What is the Great Alpine Region ?

P3 L25: Permafrost is suspected to warm at a rate of $0.04°C$ per decade at which depth ? Since when ? Does is not depend also on the ground ice content and the temperature of the permafrost (the closer is the temperature to the melting point, the larger is the latent heat consumption and the smaller is the warming rate) ?

P3 L26: Increased rock glacier velocities since the 1990s : provide a reference (Laurichard ?)

P3 L26: The increase of rock glacier velocity and some destabilization phenomena (and not their destabilization)...

P3 L27: Was really the Berard a rock glacier and not "simply" a landslide (of frozen shale and coarser debris) ?

P3 L28: Did not start the destabilization of Pierre Brune rock glacier much earlier than 1990 (see Figure 2), what is not in accordance with the sentence L26.

P3 L29: It cannot be spoken about the detachment of the active layer of the... Lou rock glacier, causing a debris flow. So far I know, there was a thunderstorm, which caused the debris flow mobilizing the active layer of the... Lou rock glacier. The permafrost table probably limited the torrential regressive erosion and consecutively the total volume of mobilized sediments.

P4 L4-6: What is the accuracy (limit of detection) of the multi-temporal orthoimagery ?

Was for instance a rock glacier moving 10 cm/y detectable as active ? How many of the 2100 rock glaciers not classified as active... could be active, to say moving more than a (few) cm/y ? This may also have an importance for the model.

P4 L13: A debris flow gully is not a rock glacier surface disturbance. It cannot be used as an indicator for rock glacier destabilization... but only for rock glacier motion (and the availability of water) in very specific topographical settings. Rock glaciers classified as destabilized on the single basis of the occurrence of a debris flow gully at their front are not and must be disregarded when building up the model.

P4 L20-21: 2 m x 2 m is quite coarse. What is accuracy (limit of detection) in a decade (2000-04 to 2012-13) ?

P5 L3 : . . . to a possible shift. . .

P5 L15: . . . Grosse Grabe and Gänder. . .

P7 L12-14: Rock glacier destabilization was observed to occur . . . at the lower limit of the permafrost zone. Is it really so ? Or what do the authors precisely mean ? Lambiel and Reynard 2001 is not here an adequate reference.

P7 L24-25: It could be worth to explain in a few words (if possible) how the PFI index is determined. Values between >0 and <1 represent the uncertainty domain of the PFI model ? Is this correct ?

P7 L30ff : The new PFI map is a shift of about 300 m of the permafrost lower limit (?), making that all PFI values within this shifting range are now set to 0, whereas in the 300 m above some are reduced to values between >0 and <1 ? Is it right ? Highest PTP values are found close to the upper boundary of the 300 m shifting range or slightly above it, no ?

P9 L12: Pixels of 2x2m or 0.5x0.5m ? How to get 0.3 m/year accuracy in the first 2000-2004 to 2008-2009 time window with 2x2m pixels ?

P9 L13: Undisturbed (instead of stable ?) active rock glaciers...

P9 L14-18: The two sentences are somewhat contradictive.

P10 L1 : The negative correlation of PISR with the destabilization probability is somewhat surprising. Is this not due to the fact that rock glaciers are (much) less frequent on southern expositions due to mountains that are not high enough to allow the occurrence of rock glaciers in such an aspect ?

P11 L29: ... reaching much more than 5-10 m/y in extreme cases of destabilization (at least seasonally) (e.g. Grabengufer – Delaloye et al. 2013, Ádjet – Eriksen et al. 2018 GRL DOI: 10.1029/2018GL077605), Jegi – Ghirlanda et al. 2016 https://media.gfz-potsdam.de/bib/ICOP/ICOP_2016_Book_of_Abstracts.pdf p.36-38, etc.)

P11 L32: See also Lambiel et al. 2008 9ICOP Proceedings

P11 L33: ... because of the high rate of sediment supply in a subjacent gully (if occurring) that may be prone to debris flow events (e.g. Kummert et al. 2017 PPP)

P12 L6 (and previous): What is a hazardous rock glacier ? This is mostly a question of connectivity toward very steep slopes or torrential gullies and transfer rate of sediments (e.g. Kummert et al. 2017 PPP), but for sure not a question of destabilization. Most of the destabilized rock glaciers are far from being hazardous (for human beings and infrastructures) ! But active "stable" rock glaciers may be.

P12 L11-13: According to my comment on P7 L30ff, it would be very interesting to explore more deeply the relationships between PTP and active rock glaciers. PFI being basically based on the front position of active rock glaciers, one can assume that migrating PFI 300 m upward would makes that the highest PTP values to be found much higher on rock glaciers... that is more likely were cracks and crevasses are located. I am wondering to what extent is this DEFROST-PTP correlation physically significant or just fortunately caused by the common morphology of rock glaciers in the French Alps ?

P12 L15ff: The comparison to the active layer detachment in the Canadian Arctic appears not to be so adequate because we are comparing two completely different phenomena/processes : shallow infiltration of water in unfrozen ground versus a deep creeping process. Moreover, the snow melt period is occurring later on northern slopes, but it starts also later. Is it so much longer ?

P13 L2 : What is this special thermal regime of rock glaciers ?

P13 L3 : Why is active layer thickening causing rock glacier destabilization ? I do not clearly understand what is meant.

P13 L6: Debris flows need debris and water. How to use their occurrence for validating the DEFROST susceptibility is so far obscure to me.

P13 L8-12: And if we look toward the future (to say again +1.5°C), what will remain "sustainable" ?

Figure 5 : Only about 25 rock glaciers are moving faster that 2 m/y in the most recent period (5% of the active ones), and not all are considered as potentially destabilized. Is this finally much or not ? About half of the potentially destabilized rock glacier (cat. 3) are moving less than 2 m/y ? I am wondering here if the criteria to define a destabilization phenomenon are all pertinent (see also my comment on Tables 1 and 3). How many rock glaciers are considered in this figure (it looks that there is only a reduced number of cat. 0 and 1) ? This could be indicated.

Figure 6 : Destabilization rating dots are almost not visible on the map.

Figure 7: PISR : I am wondering if there is not also an effect of illumination, that may make much easier to detect crevasses and cracks on a north slope (better contrast) than on an over-illuminated southern slope (less contrast) ?

Table 1 : As already said, I do not consider a debris flow gully as a sign of rock glacier destabilization. The "rugged topography" proposed by Roer et al. (2008) was related to crevasses and scarps and is not synonym of the "crack cluster" described here.

Table 3 : I am very impressed by the high number of rock glaciers displaying cracks and crack clusters. Is it due to a specific lithology ? Is it finally really a sign of destabilization ? Are all rapidly moving rock glaciers (> 2 m/y) exposing scarps and/or crevasses ? Or not ? It may be helpful to organize the table by importance of the specific disturbances as destabilization signs : crevasse(s), scarp(s), cracks cluster, crack(s). Omit gully.

---

## Author Comment (AC1) · 16 Sep 2018

**Authors' response to Anonymous Referee #1**

We wish to thank referee #1 for the valuable comments and effort put in this constructive revision. We believe that the study has significantly improved thanks to this contribution. Please find below the specific responses to each comment.

**General Comments**

*This paper uses a generalized additive model to model rock glacier stability resulting in a susceptibility map highlighting potential areas of permafrost destabilization. This is possible as the stability of rock glaciers is a reflection of the underlying permafrost stability. Results from this research are an important contribution to improving our understanding of periglacial risk and of broad significance in mountain permafrost regions. However I do have a number of comments that should be addressed before moving forward with publication.*

Overall, we agree with the concerns of the reviewer. We made few major revisions in agreement to the referees' comments:

1. The perspective of the study has been changed as we don't talk anymore about general permafrost destabilization nor degradation. We agree with the referee that rock glacier destabilization is not representative process for permafrost degradation as destabilization may have external trigger and is preconditioned by geometrical factors. The study focuses now on rock glacier destabilization and understanding these preconditioning factors. The definitions used in the manuscript have been modified in agreement to this. The manuscript title has been changed accordingly.

2. We decided to delete the section relative to the measurements of rock glacier displacement rates. The section does not fit with the study and creates confusion with the general purpose of the manuscript.

3. Debris flow gullies are not considered surface disturbances anymore as they are not linked to destabilization. Destabilization rating and susceptibility map have been updated accordingly.

4. Rock glaciers showing destabilization linked to cracks were separated from rock glaciers showing destabilization linked to crevasses and scarps. This was done to acknowledge the fact that we are not completely sure about the significance of cracks and crack clusters in the destabilization process. See P5L30 for more details.

5. Basic lithological analysis has been introduced (added a new table)

*The introduction could be improved. The relationship between rock glacier stability and permafrost destabilization needs to be more concisely and clearly presented.*

The introduction has been significantly improved. Considering the remarks from the referee #2 we decided to avoid to infer a relationship between rock glacier stability and permafrost destabilization,

as at the current state of the art we cannot support this hypothesis. The study entirely focuses on rock glacier stability now.

*Some key information about the study area is missing such as general information on ice content and active layer characteristics. There are a number of sentences that allude to its importance as observed from other papers but there is nothing specific to this paper. If this data does not exist this should be acknowledged.*

Ice content and active layer are characteristics strongly varying in rock glaciers also at the site scale. Allusion to the importance of these characteristics have been removed as they tried to relate rock glacier stability and permafrost destabilisation (see answer above).

*It may be useful to have a Data section in the Methods where you introduce the different orthoimagery datasets and their time periods, resolution and source, and, the DEM, including its resolution, source etc. I found myself having to jump around to look for this information.*

Thank you for the comment. Nevertheless, we would prefer keeping this format as data are necessary to explain the methods (e.g. orthoimages used to map destabilization). Also, since the section concerning the displacement rates has been deleted, data explanation within the sections is not redundant anymore.

*The methods explaining the  can be improved. The introduction of the GAM is ok, but it is missing some critical information. For the GAM, what type of smoothers were used and did you control for their flexibility, what were the degrees of freedom? This information is important particularly for overfitting which depends on the flexibility of a smoother, which can be controlled by the degrees of freedom. In the methods you state you will examine the accuracy of the model using sensitivity and specificity, this is not followed up on in the results.*

We would like to thank the reviewer for pointing us to this omission. The following text was inserted in section 2.3:

*"All numeric predictors were represented using spline-based smooths, for which we chose a maximum basis dimension of 4 in order to limit their flexibility and reduce overfitting. The actual degree of smoothness of the spline smooths is determined by a generalized cross-validation procedure (Wood, 2017)."*

Some degree of overfitting to the training set is of course always present, which is why independent test sets are needed for model performance estimation. We addressed this issue using spatial cross-validation (see Methods and Results). We reported the cross-validation test-set AUROC of 0.76, which is lower than the AUROC of 0.86 obtained on the training set; the latter value was not reported because it is over-optimistic.

*Some more work needs to be done to better lay out the relevance of the selected terrain variables and their connection to rock glaciers/permafrost degradation. You try to link permafrost degradation to rock glacier activity and while you do provide evidence of this through the literature the link within your own study is not clear. I assume you are using the terrain variables (i.e. PISR, slope etc) as possible proxies for surface/ subsurface conditions i.e. ice content, active layer thickening : : : etc.? Often terrain variables are selected as they serve as proxies for surface conditions that are difficult to represent through spatial datasets, however this connection is not well made throughout the methods, results or discussion.*

A more precise explanation to the relationship between rock glacier destabilization and terrain attribute has been added in section 2.3.2 and in the discussion section 4.2 :

*"The relationships withpredictor variables were found to be consistent with topographic settings observed in known cases of destabilization. Highslope angles are suggested to increase internal shear, making the landform more susceptible to destabilization (Schoeneich et5al., 2015). Convex slopes cause an extensive flow pattern as creep velocity is higher downslope the convexity (Delaloye et al.,2013). This is suggested to cause a thinning of the permafrost body and the generation of traction forces that may enhance the occurrence of surface disturbances. The PTP was found to be a significant predictor of potential destabilization. In particular, increasing potential in permafrost thaw was linked to increase susceptibility of destabilization, indicating that destabilization was more likely to occur where the permafrost zone was expected to be thawing. This seems to be consistent with the relation between destabilization and elevation, as potentially destabilized rock glacier as more often located around 2800 m.a.s.l., which roughly coincides with the lower margins of the regional permafrost zone."*

*The way you present the methods used to select the points to build the model is unclear. You say 5 randomly selected points (the size is not stated) were used to extract terrain information from each rock glacier, and that this was done as there were only 58 potentially unstable rock glaciers compared to 79 suspected destabilized rock glaciers and 119 unlikely destabilized rock glaciers. You also state that the response variable, which is representative of stable and unstable rock glacier zones were then assigned a defrost index of 1 or 0. Where those 5 points, which I assume were classified as potentially unstable, collected to increase the unstable (1's) inventory? If yes, and if you then extracted the terrain information for each of those 5 points, it's possible that the model is biased towards terrain information specific to the potentially unstable rock glaciers represented in your study. This could reducing the overall usefulness of the model when applied to the rest of landscape or perhaps lead to an overestimation of areas modeled as potentially unstable.*

Methods presentation has been improved taking into account the reviewer's remarks. Points are extracted from a point grid at 25x25 m resolution (i.e. one point per raster pixel, P7 L14). The five points are extracted from both potentially destabilized and stable/likely stable rock glaciers. Section 2.3.1:

*"Polygons of both unstable and stable areas were sampled using a 25 m x 25 m point grid in order to assign the response variable to the modelling database. The point values were then used as binary response variable with values of 0 for stable areas of (likely) stable rock glaciers, while 1 was assigned for unstable areas of potentially destabilized rock glaciers in the modelling stage."*

The size of inventory used for modelling is now stated, making clear that the model is not biased towards destabilization. Section 2.3.1:

*"Overall, the model was computed using 225 evidence of instability and 1785 evidence of stability."*

*The addition of a table outlining the size of the inventory (1's and 0's) that was used to build the model would be very helpful. This may make it easier to present the different methods i.e 5 points being extracted from unstable rock glaciers. Currently, it is not easy to get a handle on the size of the inventory and how this is integrated into the model.*

Thank you for the comment, size of the inventory is now added (see answer above)

**Specific Comments**

*P1 L3 The description of the imagery used in the abstract does not match with what is presented in the methods. In the following sections you state you have imagery from 2012-2013.*

Here it is specified that the orthoimages collection has frames from 2000 to 2013. In the methods it specified that the orthoimages belong to three main sub-period: 2000 – 2004, 2006 – 2009 and 2012 - 2013.

*P1 L6 Be more specific on the time periods being used to compare the deformation patterns.*
It is now specified that deformation patterns are observed on the "available orthoimages". Time span is presented in the pervious sentence. Abstract:

*"At first, using recent orthoimages (2000 to 2013) covering the study region, we mapped the geomorphological features that can be typically found in cases of rock glacier destabilization (e.g. crevasses and scarps). This database was then used as support tool to rate rock glaciers destabilization. The destabilization rating was assigned also taking into account the surface deformation patterns of the rock glacier, observable by comparing the available orthoimages, and the type of morphological features involved."*

*P1 L11 On P9 L31 you state that the slopes associated with higher destabilization rates is 20-40 however here in the abstract you only state that up to 30. Which is correct?*
Thank you for noticing, this has now been corrected to 25 – 30 ° (accordingly to model modifications).

*P2 L16 What methodology? No methodology is mentioned but I assume you are referring to methodology in the Sattler et al. paper? Suggest stopping sentence at ": : :degrading permafrost" or briefly layout methods of paper that show that the initiation points of debris flows weakly correlate with the spatial footprint of degrading permafrost.*
This part has been removed in the new version and the issue of "spatial footprint of degrading permafrost" is no longer treated.

*P4 L5 Additional details on the multi-temporal orthoimagery should be included i.e. dates, resolutions, source (satellite/airplane/UAV). Is this the same imagery that is introduced in section 2.2, if yes, some effort should be made to rework/better connect this information.*
This explanation has been removed as non-significant as already Marcer et al. 2017 provided to select only moving rock glaciers and in this manuscirpt data from that study are directly used.

*P4 L5 I see in the discussion you address errors associated with your mapping however, you should include a sentence in the methods explaining how you plan on assessing this.*
Yes good point, this is now stated early in the methods/ Section 2.2:

*"Nevertheless, several limitations during the mapping process were encountered, as image distortion or illumination, and will be discussed in section 4.4.1."*

*P4 L5 What features are you using to attribute a classification of active or inactive?*
Moving rock glaciers are considered as active. In Marcer et al (2017) activity is attributed by observing movements on orthoimages collections.

*P4 L6 What was the final inventory of active rock glaciers? This should be included here.*
Added the sentence to specify the inventory used and the number of active rock glaciers. Section 2.1:

*"This inventory compiled between the years 2009 – 2016 by inspecting aerialimagery in Geographical Information System (GIS)and revised by Marcer et al. (2017), revealed the high incidence of active rock glaciers in the region (i.e. 493 landforms). This inventory was used in the present study to identify active rock glaciers locations and investigate the occurrence of destabilization"*

*P4 L13 Can you be more specific or provide better descriptions of the different surface disturbances, their morphology and triggering causes. As it currently reads, it is very vague.*

Specific description of each surface disturbance is provided in Table 1 and Figure 1. This reference is now made more explicit in the text. Section 2.2

*"Surface disturbances are described in detail in Table 1 and illustrated in Figure 2."*

*P4 L 222 In the Figure 1 caption you say you used UAV images to map distinct destabilization features. But when I read the description here, there is no mention of UAV imagery. While the imagery you state you have is fairly high resolution, mapping cracks with 2 m imagery vs. mapping cracks with 50 cm imagery can be quite different. Were there resolution issues when mapping? How confident are you that you mapping all the features equally? Do you have any metrics on your mapping error? I see that you acknowledge this in your Discussion but like I mentioned in an earlier comment, a couple of sentences should be included in your methods addressing how you plan to do this.*

Thank you for pointing out this lack of clarity. UAV-retrieved hillashades were used here with the only intent to show the characteristics of the different surface disturbances. Nevertheless, we understand that this can create confusion. We therefore preferred presenting field images of the surface disturbances (new figure, now Figure 2). Concerning the mapping error, figure 2 has the intent to show how the smallest surface disturbances (i.e. cracks) look like on the orthoimages used for mapping, showing that these features can be identified. We made an error in describing the resolution of the orthoimages: older orthomosaics are at 1 x 1 m, not at 2 x 2 as stated before.  We anticipated at section 2.2 (see answer above) the upcoming discussion on the issue concerning the challenges in mapping surface disturbances.

*P4 L25 What is the biggest factor, image quality or the availability of multi-temporal images? What is the minimum resolution needed to map these features, are your images high enough resolution?*

That is quite difficult to declare in general, as the distortion may make useless images with proper lightning by creating unrealistic creeping patterns. There is a whole spectrum of difficulties that can be encountered. Nevertheless, the "methods" section may not be the best moment to start treating the subject. Difficulties concerning mapping are discussed later and now the reader is made aware of this upcoming discussion (as you suggested, see comment above). Concerning the image resolution, as specified above, a dedicated figure (2) has been added.

*P5 L5 Oblique photography? This is the first time this is mentioned, is this different than the imagery presented earlier?*

Thank you for noticing. That was a mistake, now it is removed.

*P5 L7 You finished the previous paragraph stating that you will present a slightly different definition of a destabilized rock glacier. I think the start of this paragraph should begin with a clear and concise explanation of your definition followed by the examples/ observations you use to support it. As it currently reads I do not actually know what your definition a destabilized rock glacier is. A definition needs to be presented before you describe your destabilization rating on L24.*

We agree that the text was rather confusing concerning this issue. The text has been modified in order to provide a clear definition of destabilized rock glacier already in the introduction:

*"While active rock glaciers commonly present moderate interannual velocity variations that correlate with the ground temperature (Delaloye et al., 2008; Kellerer-Pirklbauer and Kaufmann , 2012; Bodin et al., 2009), destabilized rock glaciers are characterized by a significant acceleration that can bring the*

*landform, or a part of it, to incredibly high velocities (Delaloye et al., 2013; Roer et al., 2008; Scotti et al.,2016; Lambiel, 2011; Eriksen et al., 2018). During this acceleration phase, morphological features typical of sliding processes,as crevasses and scarps, appear and grow on the rock glacier surface. This suggests that the destabilization consists of the onset of a basal sliding process over the normal creep of the rock glacier (Roer et al., 2008; Schoeneich et al., 2015). In this sense, crevasses and scarps are interpreted as the possible transition between the creep-driven and the sliding parts of the landform (Roer et al., 2008). This acceleration phase, also referred as "surge" (Schoeneich et al., 2015) or "crisis" (Delaloye et al., 2013), may last decades and it resolves in a deceleration or inactivation of the landform. Exceptionally, destabilized rock glaciers may collapse in a landslide (Bodin et al., 2016)."*

*P5 L24 Be more specific, what combination of surface disturbances/qualitative assessment of recent deformation patterns merits a rating of 1 or 2: : :?*
Specific description of each destabilization rate is provided in Table 2. This reference is now made more explicit in the text.

*P5 L29 What was specifically modelled in arctic permafrost, this description is very vague.*
It is now specified that permafrost slope failures were modelled. Section 2.3

*"The modelling followed a statistical approach similar to previous spatial prediction studies on landslides (Goetz et al., 2011) and arctic permafrost slope failures (Rudy et al., 2017) that used the Generalized Additive Model (GAM) with logistic link function"*

*P6 L5 If these 5 points were extracted from a potentially unstable rock glacier (1- presence), I assume all 5 points were represented as 1's in the model?*
We agree that the text was not clear enough and we made substantial changes. In the text it is now specified that 5 points were selected within each rock glacier perimeter. Points in unstable area belonging to potentially destabilized rock glaciers were 1's. Points in stable areas belonging to stable and likely stable rock glaciers were 0's (see answers above).

*P6 L5 What size were the multiple points that were extracted from the rock glaciers. Are you confident there is no overlap of these points? You say that model performances stabilized for more than 5 points, how was this discovered? What was the minimum and maximum number of points used? Was model performance assessed for the different number of points using only the best model?*
We are not sure to understand what it is mean by size of a point, as a point is defined only by a coordinate. Points are unique for each raster pixel and assigned a unique ID that ensures no overlap. Performance stabilization was discovered using an explorative analysis of model performance's sensitivity with respect to point sample size per rock glacier. Point sample size varied between 1 and 10 points per rock glacier. Yes that was done using only one model (i.e. with elevation instead of PTP). A more detailed description of the process that led to the choice of the sample size of 5 points per rock glaciers is now added at section 2.3.1

*"Since the rock glacier inventory counted a relatively small number of potentially destabilized cases (46 individuals), selecting only one point per rock glacier would have caused large uncertainty in the model outcome. It was therefore performed a simple exploratory analysis aimed to identify a proper amount of points per rock glacier to be used in modeling. Multiple points, from one to ten, were randomly selected within each rock glacier perimeter and used to compute a model. This was repeated ten times per each point sample size, in order to measure the variability of the model performance in relation to size of the point sample per rock glacier. Since model performances were found to stabilize for more*

*than five points selected per rock glacier, this number of points was randomly extracted per rock glacier for modelling."*

*P6 L11 Was the forward and backward stepwise variable selection not used to select the best multiple variable model (model with the lowest AIC)? The way it currently reads is that you populated a number of models with different combinations of variables, found the one with the lowest AIC and then used stepwise (forward, backward) variable selection to Identify which predictors were the best in the what would already be the best model. Suggest rewrite for clarity.*

Thank you for noticing the confusing section. Section is now made clearer by avoiding the repetition of using the AIC to find the best model. Section 2.3

*"The multiple variable models were computed using different combinations of predictor variables. Different models were compared using the Akaike Information Criterion (AIC), which Is a measure of goodness of fit that penalizes more complex models. The best multiple variable model was selected by iterating a backward-and-forward stepwise variable selection, aimed to identify which combination of predictors was better at describing the response variable by means of lower AIC."*

*P6 L18 What was the size of each cluster, how many rock glaciers in each?*

Clusters are created by dividing the dataset in five groups of equal size.

*P7 L5 Assigning your response variables a DEFROST index and then assigning a DEFROST index to your DEFROST susceptibility map decreases the clarity of inputs and outputs for the model. I would refer to your response variables as either values of 0 for stable or 1 for potentially destabilized rock glaciers*

Agree, the DEFROST acronym is now avoided (as also not fitting anymore with the aim of the study).

*P7 L8 What was your final count of 1's and 0's used to train the model? This needs to be included in the text*

Agree, count of 1s and 0s is now added (see answer above)

*P7 L11 Probability of thawing permafrost?*

We cannot define it as probability as not constructed on a statistical method.

*P7 L15 The DEM is a bit on the course side, do you think this could have affected your results? What is the average size of the rock glaciers? This general information of size, stability, mapped destabilization features should be included as a table*

Coarse DEM is not considered to have affected too much the results compared to others sources of uncertainty. Rock glaciers are large features compared to the DEM resolution. Too high resolution could be disturbed by local features of the rock glacier surface (as large boulders or crevasses) leading to non-representative values of, for example, slope and PISR. Rock glacier size in the region is not exceptional compared to others regions. Mapped destabilization features and stability are already presented in table 4.

*P7 L20 Would there be a complete absence of snow cover throughout the summer? For what periods of the year was PISR computed?*

Snow free periods are reduced to few months in summer. PISR has to be considered as proxy of several processes, involving as well snow cover duration (which is higher at low PISR values).

*P7 L21 – 34 I don't think the description of PFI should go in the Model predictor variable section. It is not actually used as an input to the model but is almost used a threshold for which you limit your resulting susceptibility map to.*
PFI is introduced here to specify how the PTP is evaluated (and used as predictor variable).

*P7 L21 It doesn't appear that you evaluate potential permafrost thaw using analytical methods presented by the others (cited in text) in this paper. You state that "Here, we used the Permafrost Favourability Index: : :.). I would remove the first part of the paragraph as it doesn't appear useful in the context of this paper. Also was the PFI recomputed specifically for this paper or are you using the results from your 2017 paper. If you are, you do not need all of the details that you include in this section.*
We did use the same method proposed by the others (cited) and applied to our data, represented by the PFI. The PTP for this region however is novel and computed for this paper specifically and therefore need to be specified with some detail. Section 2.3.2

*"The spatial distribution of degrading permafrost was evaluated following the method already presented by other studies(Hoelzle and Haeberli , 1995; Lambiel and Reynard , 2001; Damm and Felder, 2013), which consisted in artificially shifting a permafrost map proportionally to the estimated climate warming occurred between the period of validity of the map and the current climate. Here, as permafrost distribution map of the region we used the Permafrost Favourability Index (PFI) map (Marcer et al., 2017). The PFI map was calibrated using active rock glaciers as permafrost evidence and it represents the permafrost conditions during the cold episodes of the Holocene, e.g. Little Ice Age (LIA). The climate warming between the years 1850-1920 and 1995-2005 was determined using the HISTALP database (Auer et al., 2007) over the region. A permafrost distribution map was then recomputed taking into account of these temperature variations and represented the theoretical permafrost distribution in equilibrium with the current climate. By comparing this theoretical permafrost distribution and the PFI, it was obtained the Potential Thawing Permafrost zone (PTP, i.e. the so-called "melting area" in Lambiel and Reynard  (2001)). In order to use the PTP as predictor variable, it was represented by an index ranging between 0, i.e. no thaw expected, and 1, i.e. potential thaw."*

*P7 L29 What do you mean when you say the "The resulting map, which corresponded to a theoretical permafrost distribution in equilibrium with the current climate, was finally subtracted from the PFI, obtaining the Potential Thawing Permafrost zone"? In the previous sentence you say the PFI map was recomputed using the model parameters and then in this sentence you say the resulting map was subtracted from the PFI: : :? If you are subtracting two PFI maps, how are these maps different?*
The PFI map is representative of the LIA climatic conditions. The new permafrost distribution map computed by taking into account the climate warming occurring since the end of the LIA. Therefore the two maps differ as at the LIA permafrost equilibrium with climate was reached at lower elevations than nowadays. The section has been rephrased to make this concept clearer (see above).

*P8 L28 Suggest you move the sentence, "This indicates that more than 50%…." to after the next sentence where you actually provide the percentages of unstable glaciers.*
Deleted the words "this indicates" as they suggest that this sentence is a consequence of the previous one (which is not the case). Section 3.1:

*"More than 1300 surface disturbances were digitized, involving 259 active rock glaciers (Figure 6). Overall, more than the 50% of the active rock glaciers may be affected by some degree of*

*destabilization as 46 rock glaciers (9.7%) showed potential destabilization, 86 (17.0%) were suspected of destabilization and 127 (25.7%) were unlikely destabilized."*

*P9 L10 This is just personal preference but it makes the paper easier to follow if the results are presented in the same order as they are presented in the methods. Methods related to section 3.2 were the final section of the methods but here are presented as the second set of results.*
Yes, now the methods and results are in the same order.

*P9 L10 How many glaciers was this analyse done on?*
*P9 L11 If the orthoimages have resolutions ranging from 0.5 to 2 m how are displacement rates of 0.3 m/year detected and how does this correspond to _3-5 pixels? Is this possible because dates have been grouped and zero movement is inferred in the missing years. If not, how are you accounting for movement in the years you don't have imagery?*
*P9 L12 What was the limit in distorted orthophotos?*
*P9 L15 It appears as though you have grouped the first two periods, 2000 – 2004 and 2008 – 2009. This should be stated in the text. Again, how are you inferring movement for the years you do not have data? It seems like a bit of a reach to present the second period as 2009 – 2013 when you only have data from 2012 – 2013. That's a big chunk of time with no data.*
Thank you for the comments even though we deleted the related part.

*P9 L21 Unsure as to why the PTP model results are with the modelling results?*
True, sentences deleted. These general information about the PTP features are furnished already in the methods now. Section 2.3.2:

*"It is emphasized that PTP is only a proxy of permafrost degradation, which occurs at all the elevations while the PTP zone consists in a belt of 250 to 300 meters elevations that affects about 50% of the lower margins of the permafrost zone"*

*P9 L25 Earlier you mention that you also used sensitivity and specificity to assess the model performance, where are these results?*
Yes. The mention is now removed as we did not actually used them in the assessment.

*P10 L8 Why do you think the model is overestimated these areas? This should be expanded on in the Discussion section on the susceptibility model.*
Very good point, as talking about "overestimation" is incorrect. Section 3.3

*"The susceptibility predicted high destabilization susceptibility in areas belonging to stable rock glaciers."*

*P10 L18 Do you think that adding a surficial geology variable or a variable that highlights jointed bedrock would be useful?*
Good point. We added a discussion and table to highlight the relationship between destabilization rate and lithology (and yes it is useful, thank you for the comment). In section 4.1:

*"In these areas the densely jointed lithology was suspected to generate mainly pebbly rock glaciers (Matsoukaand Ikeda , 2001; Ikeda and Matsuoka , 2006). This suggested that destabilization may be more likely to develop in pebbly rock glaciers, as observed in the Berard, Roc Noir and Lou rock glaciers. Also, no rock glacier developed in crystalline lithology showed potential destabilization. However, recognizing surface disturbances on pebbly rock glaciers may be easier than in "blocky" rock glaciers, as smaller cracks are more evident. This may create a bias which should be studied more in detail."*

*P11 L 2 – 16 This is great and addresses a number of my prior questions. It is great that you acknowledge the challenges but are you able to quantitatively provide an idea of the error? Was any field validation done for any of the mapping?*

No we are not able to provide a quantitative assessment as our field validation was reduced to very few sites (mostly presented in the new Figure 2). For this reason, it is emphasized in the conclusions to include future surveys in the inventory in order to spot systematic biases or other errors.

*P11 L30 Multiclass AUROC, what do you mean here. Velocity isn't used in the model so if you evaluated other variables in a different way that should be presented in the methods and the results.*

Thank you for the comment. The section and relative issue have anyways been removed.

*P12 L8 This sections needs to be strengthened. What else to these variable tell us about process? I already made this comment in the General Comments section but I will state again that more work needs to go into explaining the importance or the terrain variables or the potential surface processes they represent.*

Thank you for the comment. A more detailed explanation has been now provided both in section 2.3.2, and in the discussion section 4.2 (see answer to general comments).

*P12 L12 Was PTP strongly correlated with the DEFROST index? At what susceptibility class was it most correlated with i.e. are areas modelled as high susceptibility the areas where the permafrost belt is expected to be thawing most?*

Spearmann correlation is equal to 0.246. The correlation is positive, indicating that areas susceptible to destabilization are found more likely in zones with high potential of permafrost thaw.

*P12 L15 Is it possible that north-facing slopes may have greater ice contents closer to the surface in part due to increased soil saturation but possibly also due to shallower active layer depths? In your area do you know how ice content varies or if there is a relationship with aspect? Also, did you look at the terrain information for your glaciers? Are there more glaciers perhaps on north-facing slopes? This would be worth looking at and presenting.*

Yes it is possible but we don't have systematic information about it. There is no available information of ice content in relation with aspect and there may not exist a correlation at all. For example, the Lou rock glacier, despite being uniformly north facing, presents very varying ice content, due to glacial/periglacial interactions mainly. Also, on this rock glacier, high ice content in areas that are not destabilized and ice-versa. Yes there are more rock glaciers on north facing slopes (low PISR), as observable in figure 7.

**Technical Corrections**
*P1 L12 Model performance should be singular*
Corrected
*P2 L13 Do you mean altitudinal?*
Sentence deleted
*P2 L9 change relationship to relation*
Corrected
*P2 L15 Suggest remove permafrost from permafrost initiation points: : :*
Sentence deleted
*P2 L23 Remove "so-called"*
Done
*P2 L24 I would suggest changing dynamical to dynamics throughout. I recognize that this term may be more common in your field but it doesn't fit in every place it is used.*

Word not used anymore

Removed (as suggested by RC2)

*P4 L3 Improve the sentence: : : "Although activity: : :"*
*P4 L4 Suggest changing the word "noticed"*

Sentences modified

Did not find the referred word

Done

*P7 L30 Suggest changing The so-called "melting area": : : to the "thawing area"*

"Melting area" refers to how it is defined in Lambiel 2001. Here, later in the study it is referred as (potentially) thawing permafrost.

*P8 L2 remove so-called, what else would the map be called?*

Removed

Done

Section removed

**Figures**

*A study area map is needed. You refer to a number of specific locations ex. P9 L1 Vanoise National Park. The study area map should include these areas to put this research into context. You could possibly use Fig 4 but this figure needs to be put into a larger geographical context as well.*

Thank you for the advice. A new figure has been added presenting the study area, cited mountain ranges and periglacial characteristics (new Figure 1).

*Figure 1 – The hillshades and orthophotos were both acquired by UAV imagery? This is the type of imagery that is available for the different temporal periods for all of France?*

UAV images are no longer presented in (now) Figure 2 as confusing. To answer your question, these were data we acquired locally to investigate geomorphometry of surface disturbances.

---

## Author Comment (AC2) · 16 Sep 2018

**Authors' response to Anonymous Referee #2**

We wish to thank referee #2 for the valuable comments and effort put in this constructive revision. We believe that the study has significantly improved thanks to this contribution. Please find below the specific responses to each comment.

**General comments**:

*Good paper on an subject gaining in importance in the understanding of the behavior (stability) of mountain debris slopes underlain by permafrost : the so-called destabilization of rock glaciers. On the basis of an extended original dataset the authors spatially model the susceptibility of a location to be affected by a destabilization process in the French Alps. The preparatory work of inventorying the destabilization indices on rock glaciers in the entire French Alps is impressive and constitutes for me the most attractive part of the paper. The statistical modelling approach appears to be good, but can only provides results which are very difficult to validate in my point of view. It is so partly less convincing. Some of the terminology used in the paper (permafrost destabilization, rock glacier destabilization, stable/unstable rock glacier, hazardous rock glacier) is somewhat unclear and even questionable. It has to be checked carefully all the paper along. I also have some questions about the interpretation/use of some the destabilization indices. Maybe some of the results may change in accordance. After having been revised the paper will be definitely very worth of being published in TC. I hope my comments/suggestions can be useful for the authors.*

Overall, we agree with the concerns of the reviewer. We made few major revisions in agreement to the referees' comments:

1. The perspective of the study has been changed as we don't talk anymore about general permafrost destabilization nor degradation. We agree with the referee that rock glacier destabilization is not representative process for permafrost degradation as destabilization may have external trigger and is preconditioned by geometrical factors. The study focuses now on rock glacier destabilization and understanding these preconditioning factors. The definitions used in the manuscript have been modified in agreement to this. The manuscript title has been changed accordingly.

2. We decided to delete the section relative to the measurements of rock glacier displacement rates. The section does not fit with the study and creates confusion with the general purpose of the manuscript.

3. Debris flow gullies are not considered surface disturbances anymore as they are not linked to destabilization. Destabilization rating and susceptibility map have been updated accordingly.

4. Rock glaciers showing destabilization linked to cracks were separated from rock glaciers showing destabilization linked to crevasses and scarps. This was done to acknowledge the fact that we are not completely sure about the significance of cracks and crack clusters in the destabilization process.

5. Basic lithological analysis has been introduced

**Specific comments:**

*Title is not good. "Inferring the destabilization susceptibility of mountain permafrost: : :" has no real sense, permafrost being a thermal phenomenon. This is not the scope of the paper, which is conversely dealing with the mechanical "destabilization" of rock glaciers. There is however a much better alternative proposed by the authors on P3 L15, which can give a title like "Evaluating the destabilization susceptibility of active rock glaciers in the French Alps".*

Agree. Title changed to "Evaluating the destabilization susceptibility of active rock glaciers in the French Alps"

*Abstract : may have to be adapted after revision of the paper*
Abstract adapted to the revisions

*P1 L17 Express what is meant precisely by (widespread) permafrost degradation. This is not clear at all, but a very important concept for this paper. Check then in the whole paper if the concept is used always exactly with the same sense. Prefer "Permafrost has shown signs of w. degradation for the past decades in the European Alps".*
A better definition and explanation of permafrost degradation is now provided in the introduction:

*"Warmer mean annual air temperatures (IPCC , 2013) are linked to a general trend of increasing permafrost temperature (e.g.Harris et al., 2003) and water content (e.g. Ikeda et al., 2008) causing permafrost degradation, a phenomenon widely observed inthe European Alps (Haeberli et al., 1993, 2010; Springman et al., 2013; Bodin et al., 2015). Permafrost degradation occurrence is dependent on the ground properties, snow cover interactions and permafrost ice content (Scherler et al., 2013) and is therefore an heterogeneous phenomenon. Permafrost grounds affected by degradation experience a loss in stiffness due to the increasing ice ductility and reduced internal friction caused by the warmer ice and increasing water content (Davies et al. , 2001; Haeberli et al., 1997; Harris and Davies , 2001; Nater et al., 2008; Huggel et al., 2010).Abnormal rockfall activity at high elevations (e.g.Ravanel and Deline , 2010) and increasing rock glaciers displacement rates (Delaloye et al., 2008) are indicators of this changeof state in the mountain permafrost"*

*P1 L18 : The connection between air temperature and ground temperature is tricky (snow buffering effect) and I do not really understand the meaning of "extreme warm air temperature" in the sentence. I would suggest to simplify it as "Warmer climatic conditions are expected to cause: : :" (eventually is a redundancy and can be omitted). Is there not more recent and more adapted references ? Finally, I do not see the link with the previous sentence.*
It is now acknowledged that ground properties and snow cover have a significant impact on the connection between air temperature and ground temperature (see above). "Extreme warm air temperature" omitted. The whole section has been re-arranged since, as you suggested, there was poor connection between sentences.

*P1 L20 : The "thermal inertia" is in particular highly related to the ice content, which can be relatively high in a rock glacier. This should be mentioned. Ground instead of soil.*
This is now acknowledged (see above), thank you for the advice.

*P1 L21 : I do not understand the meaning of this part of the sentence in the context of the present study. Why currently ? Is really the reference adequate ? Would not be for instance Scherler et al. 2013 (https://doi.org/10.1002/jgrf.20069, see in particular fig. 5, where the modelled impact of climate warming on two sites with very contrasted ice content is illustrated) more appropriate ?*
The sentence has been removed as not meaningful. Thank you for the reference.

*P2 L1 : What kind of "other processes" ?*
Now omitted.

*P2 L3 : Rock glacier destabilization can be caused by other factors than only climate induced permafrost warming (e.g. cited Roer et al. 2008, Delaloye et al. 2013). The sentence must be adapted in consequence. This is a very important point, because the susceptibility model appears to be based on the assumption of a climate impact only.*
Yes, that's a very crucial point and thank you for the comment. The manuscript has been adapted to this comment by acknowledging that destabilization may be triggered by different factors i.e. mechanical and climatic). Nevertheless, the occurrence of destabilization is finally discriminated by the landform predisposition to destabilization, i.e. the "geometrical factors" (Delaloye et al, 2013). This is a very important point which allows to make clear through the paper that:

1. The modelling part aims to investigate the predisposition to rock glacier destabilization only
2. Testing the significance of the PTP in the predisposition means testing the hypothesis that rock glaciers located at the lower margins of the permafrost zone are more susceptible to destabilization.

Modifications are made through the text to adapt the study to these two concepts. Specifically, your point is acknowledged in the introduction:

*"An overload on the glacier surface caused by a landslide or glacio-isostatic uplift can cause a compressive wave that propagates through the landform increasing its displacement rates and consequent destabilization (Delaloye et al., 2013; Roer et al., 2008)."*

*P2 L4 : Delaloye et al. 2013 (and not 2008). To be changed also further in the paper.*
Thank you for noticing it.

*P2 L4-5 : These events are far from all "representing a serious threat for alpine communities". Sentence to be adapted.*
Sentence (and general paper "tone") has been adapted. In introduction:

*"[..] destabilization and increased displacement rates may precondition significant mass movements that in particular topographic setting may represent an hazard (Kummert and Delaloye , 2018)"*

*P2 L5 (and many times in the paper): Permafrost destabilization. What is this ? The authors are rather talking about the destabilization of frozen ground inducing almost significant mass movements (>100'000m3 ?). Permafrost destabilization appears to be an inadequate terminology that must be replaced by rock glacier or debris slope destabilization and adapted all the paper along.*
Good point. Permafrost destabilization has been replaced by "rock glacier destabilization" or "creeping permafrost destabilization". Also, through the manuscript is made clear that we specifically investigate rock glacier destabilization only. In Introduction:

*"The purpose of this study was to obtain regional-scale insights into the issue of destabilizing rock glaciers in the French Alps."*

*P2 L7, L9-13: Permafrost degradation. Again, what are we talking about ? About a complete ice melt = permafrost has disappeared, the temperature is now above freezing point ? Or about an increased liquid water content (partial ice melt) by warmer permafrost temperature (without any permafrost thaw) ? Is a permafrost warming from -1 to -0.5_C consecutively increasing the liquid water content a permafrost degradation ? In my point of view yes, making that almost all permafrost in the Alps (and in many places elsewhere) is currently degrading ! L13 : It looks that permafrost degradation is here*

*considered as where permafrost is still occurring where it should not be (that is, despite current ground surface thermal conditions that could no more permit its occurrence) ?*

Good point. It is now acknowledged in the paper that permafrost degradation occurs everywhere in the Alps (see introduction above). It was incorrect to blindly put on the same level degrading permafrost and permafrost still occurring despite unsuitable thermal conditions. This distinction is now made clearer, emphasizing that the PTP is only a proxy of permafrost degradation under the assumption that permafrost at lower elevations is temperate, richer in water and more sensitive to climate variations. Section 2.3.2

*"PTP is used under the hypothesis that degradation is more intense at the lower margins of the permafrost zone as permafrost may be temperate, richer in water and more sensitive to climate variations"*

*P2 L14 : What are stable and unstable rock glaciers ? In addition, I cannot agree with the sentence, which seems to be based on the assumptions that all rock glacier destabilizations are induced by climate warming (permafrost warming) and that all rock glaciers with "degrading" permafrost conditions have to destabilize.*

Yes, misleading and inaccurate sentence, now it has been completely rephrased. Difference between stable and unstable rock glacier is defined in the introduction:

*"While active rock glaciers commonly present moderate interannual velocity variations that correlate with the ground temperature (Delaloye et al., 2008; Kellerer-Pirklbauer and Kaufmann , 2012; Bodin et al., 2009), destabilized rock glaciers are characterized by a significant acceleration that can bring the landform, or a part of it, to incredibly high velocities (Delaloye et al., 2013; Roer et al., 2008; Scotti et al.,2016; Lambiel, 2011; Eriksen et al., 2018)."*

It is now clarified that increasing temperatures and (possibly) degrading permafrost may trigger destabilisation if the rock glacier geometry allows it (i.e. rock glaciers on flat topography will go towards inactivation if degrading). It is also explained that there is a large number of observation that recognized external factors as destabilization triggers P2 L17 -19. Introduction:

*"Nevertheless, not all rock glaciers experiencing permafrost degradation or mechanical overload are, or will be, destabilized. Permafrost degradation generally causes permafrost thaw in the landform and consequent inactivation (Scapozza et al., 2010). Destabilization was observed only in rock glaciers presenting a topographical predisposition to mass movements, as steep slopes and flow across a convex section (Roer et al., 2008; Delaloye et al., 2013). This suggests that there is a terrain predisposition of the rock glaciers to the onset of a destabilization phase."*

*P2 L20: "Observing rock glacier dynamics and morphology can be rather useful" for what ?*
Deleted. Not relevant in the context as the study.

*P2 L21: Permafrost degradation (complete ice melt ?) in ice-rich landforms does not directly cause the mobilization of significant amount of materials. It makes the material easily erodible, but does not put it in motion. It does not trigger debris flow, but only precondition it. Moreover, this is the (increased) motion of the rock glacier, which is making (more) materials available for later debris flow events (e.g. Kummert)*
It is now acknowledged that rock glacier destabilization may precondition mass movement if the landform is located in particular topographical settings. Introduction:

*"[..] destabilization and increased displacement rates may precondition significant mass movements that in particular topographic setting may represent an hazard (Kummert and Delaloye , 2018)"*

*P2 L23 : An increase of the liquid water content is assumed to cause the so-called destabilization (and not can).*
Sentence modified. Introduction:

*"Warmer climate and linked permafrost degradation on the other hand, its assumed to cause an increase of water content in the permafrost body and the onset of water saturated shear layers where sliding may occur, possibly triggering the crisis"*

*P2 L24-29: About the occurrence of destabilization of active rock glaciers, see also Lambiel et al. 2008. Proceedings of the Ninth International Conference on Permafrost, Fairbanks, Alaska, 1 pp. 1019-1025, in particular Table 2 and related text.*
We are sorry but we couldn't find this reference in the proceedings document.

*P2 L26 : : : : exceptionally (instead of eventually ?) lead to the collapse of the rock glacier (or a significant part of it)*
Yes, it is now specified that rock glacier collapse is an exceptional event.

*P2 L29: Lambiel and Reynard (2001) has nothing to do with destabilization*
True, wrong reference.

*P3 L13: Is DEFROST the most appropriate name for the model, because it helps to evaluate the destabilization susceptibility of active rock glaciers only, and not permafrost (or all permafrost slopes)?*
True. Although we were very proud of this beautiful acronym, we decided to not use it here as not pertinent. It is now referred to rock glacier destabilization susceptibility.

*P3 L19 : Sorry but 15'000 km2 fits with the total area of the French Alps (50-75 x 250 km) and consequently not with the area above 1500 m. And why to mention this latter area?*
True, we must have made a mistake. As you suggest, useless sentence, now omitted.

*P3 L22: Climate is changing fast. Indicate the reference time period for the elevation values of the annual 0_ isotherm.*
Added reference period (and corrected actually wrong values)

*P3 L23: What is the Great Alpine Region ?*
The European Alps as defined in some climatology papers but simply "European Alps" will be better.

*P3 L25: Permafrost is suspected to warm at a rate of 0.04_C per decade at which depth ? Since when ? Does is not depend also on the ground ice content and the temperature of the permafrost (the closer is the temperature to the melting point, the larger is the latent heat consumption and the smaller is the warming rate) ?*
Details about location and time period added. Section 2.1:

*"The only deep permafrost borehole in the region, located in the Ecrins massif in temperate permafrost (-1.3°C) with low ice content, showed a temperature increase rate of 0.04°C perdecade between 2010 and 2014 (Schoeneich et al., 2012), similarly to many sites in Switzerland where data series are longer(PERMOS , 2016) ."*

*P3 L26: Increased rock glacier velocities since the 1990s : provide a reference (Laurichard?)*
Yes, reference added.

*P3 L26: The increase of rock glacier velocity and some destabilization phenomena (and not their destabilization): : :*
Agree, corrected

*P3 L27: Was really the Berard a rock glacier and not "simply" a landslide (of frozen shale and coarser debris)?*

In the Berard site it was observed creeping and massive ice of (probably) periglacial genesis, features that brought Bodin et al (2016) to define it as rock glacier. We would like to stick to the definition proposed by previous authors. Nevertheless, we may not fully understand your concerns about the definition of the Berard site and its implications with this study. Assuming that you are concerned by the exceptionality of the event, we changed the sentence in section 2.1:

*"In 2006 the Berard rock glacier collapsed causing a landslide of 250 000 m3, a very exceptional event that was possibly linked to the rare characteristics of this site, e.g. uncommonly fine grained debris (Bodin et al., 2016)"*

Also, in Figure 8, the Berard example has been replaced with a "more conventional" example of destabilization (Iseran destabilized rock glacier).

*P3 L28: Did not start the destabilization of Pierre Brune rock glacier much earlier than 1990 (see Figure 2), what is not in accordance with the sentence L26.*

Pierre Brune was showing a crevasse since the 70s. Nevertheless, surface velocity were very low until the 90s. The crisis occurred mainly at the end of the 90s (velocities up to 5 m/s) and currently ongoing.

*P3 L29: It cannot be spoken about the detachment of the active layer of the: : : Lou rock glacier, causing a debris flow. So far I know, there was a thunderstorm, which caused the debris flow mobilizing the active layer of the: : : Lou rock glacier. The permafrost table probably limited the torrential regressive erosion and consecutively the total volume of mobilized sediments.*

Yes, the Lou frontal slides were recognized to be concentrated flow phenomena (Kummert et al, 2017) only after the submission of this study. We agree that we cannot talk about permafrost degradation/rock glacier destabilization as a trigger for that event.

*P4 L4-6: What is the accuracy (limit of detection) of the multi-temporal orthoimagery ? Was for instance a rock glacier moving 10 cm/y detectable as active ? How many of the 2100 rock glaciers not classified as active: : : could be active, to say moving more than a (few) cm/y ? This may also have an importance for the model.*

This part was removed as it was actually already treated in Marcer et al (2017). To answer you question I quote that study: *"Also, due to the relatively short time span of 8–15 years covered by the aerial imagery, the movement of rock glaciers creeping at small velocities (~0.1–0.2 m/y) may have remained undetected."*

*P4 L13: A debris flow gully is not a rock glacier surface disturbance. It cannot be used as an indicator for rock glacier destabilization: : : but only for rock glacier motion (and the availability of water) in very specific topographical settings. Rock glaciers classified as destabilized on the single basis of the occurrence of a debris flow gully at their front are not and must be disregarded when building up the model.*

Agree. This was a misjudgement due to the (over)interpretation of the Lou event. The debris flow gullies are now disregarded in this study and the destabilization rating and model computation updated consequently.

*P4 L20-21: 2 m x 2 m is quite coarse. What is accuracy (limit of detection) in a decade (2000-04 to 2012-13) ?*

Wrong value, it is actually 1 x 1 m (or finer according to location). Apart this, section removed.

*P5 L3 : : : : to a possible shift: : :*

Corrected

*P5 L15: : : : Grosse Grabe and Gänder: : :*
Corrected

*P7 L12-14: Rock glacier destabilization was observed to occur : : : at the lower limit of the permafrost zone. Is it really so ? Or what do the authors precisely mean ? Lambiel and Reynard 2001 is not here an adequate reference.*
Inadequate sentence. This may not be true as only few studies actually report this information (e.g. Scotti et al, 2016; Bodin et al, 2016, personal knowledge on Pierre Brune and Roc Noir rock glaciers). This is now omitted (as the inadequate reference).

*P7 L24-25: It could be worth to explain in a few words (if possible) how the PFI index is determined. Values between >0 and <1 represent the uncertainty domain of the PFI model ? Is this correct ?*
It is now added that the PFI is based on rock glacier inventory. The PFI varies between 0 (climate favourable to the existence of a relict rock glacier) and 1 (climate favourable to the existence of a rooting zone of an active rock glacier). Nevertheless, we now avoid going into the details of the meaning of the indexes, as it can result too confusing and complex as explanations. PTP description is now more qualitative and only the PTP index is explained.

*P7 L30ff : The new PFI map is a shift of about 300 m of the permafrost lower limit (?), making that all PFI values within this shifting range are now set to 0, whereas in the 300 m above some are reduced to values between >0 and <1 ? Is it right ? Highest PTP values are found close to the upper boundary of the 300 m shifting range or slightly above it, no ?*
No, the highest PTP values are found where PFI was equal 1 during the LIA and equal to zero in the present climatic conditions. In other words, we expect thaw where there was permafrost during the LIA and now it is not supposed to hold in the current climate. This confusion is probably due to the fact that the section was poorly explained, involving too many indexes and complexity. As explained above, section has been described more qualitatively, hopefully making it more clear. Section 2.3.2:

*"The spatial distribution of degrading permafrost was evaluated following the method already presented by other studies(Hoelzle and Haeberli , 1995; Lambiel and Reynard , 2001; Damm and Felder , 2013), which consisted in artificially shifting a permafrost map proportionally to the estimated climate warming occurred between the period of validity of the map and the current climate. Here, as permafrost distribution map of the region we used the Permafrost Favourability Index (PFI) map(Marcer et al., 2017). The PFI map was calibrated using active rock glaciers as permafrost eviedence and it represents thepermafrost conditions during the cold episodes of the Holocene, e.g. Little Ice Age (LIA). The climate warming between the years 1850-1920 and 1995-2005 was determined using the HISTALP database (Auer et al., 2007) over the region. A permafrost distribution map was then recomputed taking into account of these temperature variations and represented the theoretical permafrost distribution in equilibrium with the current climate. By comparing this theoretical permafrost distribution and the PFI,it was obtained the Potential Thawing Permafrost zone (PTP, i.e. the so-called "melting area" in Lambiel and Reynard (2001)). In order to use the PTP as predictor variable, it was represented by an index ranging between 0, i.e. no thaw expected, and 1, i.e. potential thaw."*

*P9 L12: Pixels of 2x2m or 0.5x0.5m ? How to get 0.3 m/year accuracy in the first 2000-2004 to 2008-2009 time window with 2x2m pixels ?*
Same error as above, it actually is 1 x 1m.

*P9 L13: Undistorted (instead of stable ?) active rock glaciers: : :*
We prefer keeping the stability scale as we are talking about rock glaciers destabilization and not only of presence/absence of surface disturbances.

*P9 L14-18: The two sentences are somewhat contradictive.*
Section deleted.

*P10 L1 : The negative correlation of PISR with the destabilization probability is somewhat surprising. Is this not due to the fact that rock glaciers are (much) less frequent on southern expositions due to mountains that are not high enough to allow the occurrence of rock glaciers in such an aspect ?*
We are not sure if we can offer a convincing explanation for this phenomenon at this point. In the Discussion (section 4.3) we point the reader to the importance of water in causing the destabilization of rock glaciers (Ikeda et al., 2008).

Concerning the second part of the reviewer's comment it is important to distinguish between (1) the probability of a specific location presenting a destabilized rock glacier, and (2) the probability that a given rock glacier shows signs of destabilization. Clearly, the reviewer refers to the probability of type (1), which partly relates to rock glacier occurrence per se. This paper, however, only addresses probability type (2), which is conditional on the occurrence of a rock glacier, and therefore unrelated to the question where rock glaciers are more frequent. In other words, yes we have more active rock glaciers on northern expositions than southern, but still the probability of having destabilization is proportionally higher in northern expositions than in southern expositions.

*P11 L29: : : : reaching much more than 5-10 m/y in extreme cases of destabilization (at least seasonally) (e.g. Grabengufer – Delaloye et al. 2013, Ádjet – Eriksen et al. 2018 GRL DOI: 10.1029/2018GL077605), Jegi – Ghirlanda et al. 2016 https://media.gfzpotsdam. de/bib/ICOP/ICOP_2016_Book_of_Abstracts.pdf p.36-38, etc.)*
Thank you for the reference, now integrated in the study.

*P11 L32: See also Lambiel et al. 2008 9ICOP Proceedings*
Sorry but we could not find the reference you proposed.

*P11 L33: : : : because of the high rate of sediment supply in a subjacent gully (if occurring) that may be prone to debris flow events (e.g. Kummert et al. 2017 PPP)*
True (although section removed)

*P12 L6 (and previous): What is a hazardous rock glacier? This is mostly a question of connectivity toward very steep slopes or torrential gullies and transfer rate of sediments (e.g. Kummert et al. 2017 PPP), but for sure not a question of destabilization. Most of the destabilized rock glaciers are far from being hazardous (for human beings and infrastructures) ! But active "stable" rock glaciers may be.*
Yes, it is now made clearer through the manuscript that hazard is discriminated by connectivity (see comment above)

*P12 L11-13: According to my comment on P7 L30ff, it would be very interesting to explore more deeply the relationships between PTP and active rock glaciers. PFI being basically based on the front position of active rock glaciers, one can assume that migrating PFI 300 m upward would makes that the highest PTP values to be found much higher on rock glaciers: : : that is more likely were cracks and crevasses are located. I am wondering to what extent is this DEFROST-PTP correlation physically significant or just fortunately caused by the common morphology of rock glaciers in the French Alps ?*
This question is very similar to P10 L1 and we address it in the same way. As you say, it is true that, due to the method used to produce the PTP map, many active rock glaciers present a high PTP index. Nevertheless, the proportion of destabilization VS stability is still higher for higher PTP than for lower PTP. Significance of the PTP as predictor in the model indicates that there is indeed a significant correlation between destabilization and this predictor.

*P12 L15ff: The comparison to the active layer detachment in the Canadian Arctic appears not to be so adequate because we are comparing two completely different phenomena/processes : shallow*

*infiltration of water in unfrozen ground versus a deep creeping process. Moreover, the snow melt period is occurring later on northern slopes, but it starts also later. Is it so much longer ?*

The comparison is now avoided. This is a good question and we cannot provide an answer. We are suggesting that between north exposed slopes and south exposed slopes there is a strong variability in snow cover duration. Considering the impact of snow cover on permafrost, we suggest to investigate that phenomena to explain the fact that most of the destabilisation occurs at low solar radiation.

*P13 L2 : What is this special thermal regime of rock glaciers ?*
*P13 L3 : Why is active layer thickening causing rock glacier destabilization ? I do not clearly understand what is meant.*
*P13 L6: Debris flows need debris and water. How to use their occurrence for validating the DEFROST susceptibility is so far obscure to me.*

Inappropriate section, deleted in agreement to a focus to rock glacier destabilization only.

*P13 L8-12: And if we look toward the future (to say again +1.5_C), what will remain "sustainable" ?*

We cannot provide a correct answer to this question in the context of rock glacier destabilization.

Concerning, the PTP which can be extrapolated using future climatic scenarios, an increase of + 1.5 with respect to present levels (i.e. + 3 since the preindustrial), will result in a shift of ~500m upslope of the lower limits of the permafrost zone.

*Figure 5 : Only about 25 rock glaciers are moving faster that 2 m/y in the most recent period (5% of the active ones), and not all are considered as potentially destabilized. Is this finally much or not ? About half of the potentially destabilized rock glacier (cat. 3) are moving less than 2 m/y ? I am wondering here if the criteria to define a destabilization phenomenon are all pertinent (see also my comment on Tables 1 and 3). How many rock glaciers are considered in this figure (it looks that there is only a reduced number of cat. 0 and 1) ? This could be indicated.*

Considering that this figure (and relative section) has been removed, please find our answer to the issue of criteria of destabilization at "Table 3" comment.

*Figure 6 : Destabilization rating dots are almost not visible on the map.*

New version of the figure proposed

*Figure 7: PISR : I am wondering if there is not also an effect of illumination, that may make much easier to detect crevasses and cracks on a north slope (better contrast) than on an over-illuminated southern slope (less contrast) ?*

We checked if there was a systematic issue with illumination and we did not find any. It is true that strong illumination makes surfaces featureless. However, this issue exists in all aspects and does not concerns all the orthoimages.

*Table 1 : As already said, I do not consider a debris flow gully as a sign of rock glacier destabilization. The "rugged topography" proposed by Roer et al. (2008) was related to crevasses and scarps and is not synonym of the "crack cluster" described here.*

Yes it has been now removed. Thank you for the "rugged topography" clarification.

*Table 3 : I am very impressed by the high number of rock glaciers displaying cracks and crack clusters. Is it due to a specific lithology ? Is it finally really a sign of destabilization ? Are all rapidly moving rock glaciers (> 2 m/y) exposing scarps and/or crevasses ? Or not ? It may be helpful to organize the table by importance of the specific disturbances as destabilization signs : crevasse(s), scarp(s), cracks cluster, crack(s). Omit gully.*

Very good point, the manuscript has been majorly changed following this. Cracks and crack clusters are surface disturbances that could be observed on the field in two cases of known destabilisation, therefore we consider them as a destabilisation evidence. Still, we agree there is a lot of uncertainty

about this features and their significance as they are very common. We acknowledge this in the text now and potentially destabilised rock glaciers (cat 3) were separated into two different categories according to the surface disturbances they were showing. Rock glaciers showing scarps and crevasses as major evidence of destabilisation were classified into the cat 3a, while in 3b were classified rock glaciers showing crack clusters. Section 2.2.1

*"Potentially destabilized rock glaciers were ultimately classified into two different categories according to the type of surface disturbances observed. Most of the destabilization cases observed by previous studies described rock glaciers characterized by surface disturbances that may reach several meters of depth, i.e. crevasses and scarps, and therefore suggested to split the permafrost body. These surface disturbances can be observed in coarse grained (i.e. blocky, sensu Ikeda and Matsuoka (2006)) rock glaciers. Nevertheless, in the French Alps many active rock glaciers are fine grained and some destabilization cases, e.g the Lou (Schoeneich et al., 2017) and Iseran (Serrano , 2017) rock glaciers, were observed to be characterized by the presence of cracks only. These surface disturbances are shallower than crevasses and scarps and therefore suggested to affect only the upper layer of the rock glacier. As these observations were relatively recent, at present there is still not enough knowledge concerning the significance of these shallow cracks in the context of rock glaciers destabilization. We therefore decided to separate rock glaciers showing shallow surface disturbances from rock glaciers showing deep surfaces disturbances into two5distinct classes in order to make the reader aware of this gap in knowledge"*

To answer your questions regarding rock glacier velocity (although not in the text anymore): not all rapidly moving rock glacier show crack or crevasses and not all cat 3 rock glaciers showing crack or crevasses are rapidly moving. Rock glaciers showing only crack clusters may be rapidly moving (one site was observed on orthoimages to move at more than 6 m/y over 5 years, while the Lou currently moves at more than 3.5 m/y). This is nevertheless a very interesting subject, that will hopefully be developed more into details in a future study.

---

## Author Response (AR1)

**Authors' response to the Editor**

Dear Editor,

We are pleased to submit you the new version of our manuscript. We believe that the study has greatly improved thanks to the referees comments and we are very glad to have received such constructive review. We have addressed all the point raised by the referees, for a detailed account, please consult can find the authors' response to the referees in the discussion forum.

In addition to that we would like to point out several editorial modifications that may be of your interest:

1. The title has changed to "Evaluating the destabilization susceptibility of active rock glaciers in the French Alps". This was done in agreement to the reviews of referee #2 that questioned the significance of rock glacier destabilization in the context of degrading permafrost. It was decided to focus the study on the rock glacier destabilization phenomena only.
2. The sections relative to the estimations of rock glaciers displacement rates have been removed. This was done as this part of the study resulted in confusion and did not bring relevant knowledge to the subject.
3. Figures were subjected to major changes. Also, a new figure (now Figure 1) describing the study area has been added.
4. A new table (now Table 3) describing the relation between rock glacier destabilization and lithology has been added.

We hope you will consider our manuscript relevant for TC Discussions and we are looking forward to your feedback.

Kindly,

Marcer Marco

[revised manuscript text omitted]

---

## Author Response (AR2)

**Evaluating the stabilization susceptibility of active rock glaciers in the French Alps**

Marcer et al

*The paper has improved since its initial submission. I thank the authors for their very detailed response to my reviews. I have some additional comments that are primarily focused on the introduction and methods. There are also a number of spelling and grammatical errors throughout that should be addressed. I have pointed out a number of them but not all.*

We thank the reviewer for this valuable revision of our manuscript. We are glad that previous changes were appreciated and we are glad of this new revision. We agree with most of the comments provided. The manuscript has surely improved. Our English native speaker co-author did a detailed revision of grammar and spelling mistakes.

*Abstract*

*The abstract is much improved, it is clear and concise.*

*P1L4 … as a support tool…*

Corrected

*P1L11 …model performance was good…*

Corrected

*P1L19 suggest: The occurrence of permafrost degradation is dependent on the ground properties, snow cover interactions and ground ice content () and is therefore a heterogeneous phenomenon.*

Thank you for the suggestion, text modified accordingly

*P1L20 suggest: loss in strength*

Accepted suggestion.

*P2L3 Therefore, there is a growing…*

Corrected

*P2L4 … to allow for a targeted risk…*

Corrected

*P2L11 "…destabilization consists on the onset"? Suggest you reword this sentence*

Sentence modified to (P2L11):

*"This suggests that the destabilization occurrence is caused by a basal sliding process over the normal creep of the rock glacier"*

*P2L13 creep-driven sections and sliding sections*

Corrected

*P2L13 Change the word exceptionally, doesn't make sense here*

Changed to *"In very rare circumstances"*

*L17 Reword sentence, doesn't work as is*

Thank you for noticing. The sentence was not well located. Now we changed to a topic sentence to introduce the paragraph (P2L18):

*"The destabilization process can be triggered either by a mechanical forces or changes in climate"*

*L20 … leading to destabilization…*

Corrected

*L21 Sentence does not flow, reword to strengthen the point*

Changed to (P2L19-21):

*"A warmer climate may also trigger a destabilization crisis as increasing temperatures may cause permafrost degradation of the rock glacier. This process may result in the onset of water saturated shear layers where sliding occurs, triggering the crisis"*

*L25 – 29 I do not understand what you are trying to present in these sentences.*

The sense of the sentences is to point out that the terrain predisposition to destabilization occurrence is fundamental. The sentences are now changed (P2L25-29):

*"Although triggers are necessary to the destabilization occurrence, not all rock glaciers subjected to these external forces destabilize. For example, permafrost degradation in rock glaciers mainly causes permafrost thaw and results in inactivation (Scapozza et al., 2010). Destabilization can be triggered only if there is a local topographical predisposition of the rock glacier to this process, such as steep slopes (Roer et al., 2008; Delaloye et al., 2013). Therefore, the terrain attributes of the rock glaciers to the onset of a destabilization phase are a critical parameter in the process occurrence."*

*Last paragraph of the introduction should be cleaned up to make the point clearer. Also the introduction also does not include any details on the type of model being used. This should be included and should be mentioned in the abstract as well.*

This paragraph has been rewritten and point is made more clear. The GAM is now mentioned in the introduction (P3L9)

*P3L25 XX century?*

Corrected

*L27 ..was compiled…*

Corrected

*L32 20th century has something off with the superscript*

Corrected

*P4L2 …similar to …*

Corrected

*L4 Provide context for the Laurichard Rock Glacier, where is it? Linked to increasing temperature? Be direct, is this the phenomenon you are talking about?*

Sentence modified accordingly (P4L4-5):

*"Increasing air temperature was also addressed to be responsible for the acceleration since the late 1990s of the active Laurichard rock glacier located in the Combeynot massif of the French Alps (Bodin et al., 2009)."*

*L4 Remove sentence…at the same time….*

Removed

*L3-11 I don't see how the specifics of these examples are pertinent to your study. I would suggest stating that there have been a number of important changes associated with rock glaciers i.e. destabilization due to increasing temperatures, increases in the development of crevasses, and that these changes can/have led to risks in a number of regions. Summarize these examples more effectively.*

These examples were required by RC1, as it was requested to specify the link between permafrost and climate change in the region. We nevertheless accept the comment and presented the paragraph in a more succinct form. Please find the new version of the paragraph at P3L32 – P4L9

*L33 You state you used all three available orthoimages to ensure that you were mapping surface disturbances and not just compression features. Due to the large temporal gaps how does mapping from the three mosaics help? Are you using the temporal images to identify if new crevasses have formed and hence a surface disturbance? Please clarify this section.*

In this section is described the surface disturbances mapping. Here, multiple orthoimages are used to overcome the challenges of using a single image in mapping (as snow cover, bad lightning). Observing the evolution of surface disturbances is a methodology for the destabilization rating.

*P5L7 recurrence of features on destabilized rock glaciers…*

"Recurrent" was deleted as meaningless and misleading in this sentence.

*L8-29 This section presents too many examples without clearly laying out why they are presented. How does past analysis of rock glaciers help you assign destabilization ratings? How did you use the three temporal periods and changes you might have mapped when assigning ratings? This is the type of information that should be highlighted. One or two past examples that you may have used to inform how you interpreted changes is fine but this sections needs to be much more concisely written*

We agree that these paragraph were not properly presented. Please note the substantial revision at P5L6-27

To answer the questions:

This section is needed to define how we made the link between destabilization and features observable on multiple images PL6:

*"Assigning a rating to quantify the degree of destabilization of a rock glacier required the definition of the characteristics of the "typical" destabilized rock glacier that can be observed on multiple orthoimages. To do so, we investigated the features of destabilized rock glaciers reported in the literature that could be observed by orthoimagery interpretation"*

The use of the three orthoimages is now better defined at P5L26 :

*"A comparison of the available IGN multi-year orthoimagery was used to observe the temporal evolution of the surface disturbances and surface deformation patterns."*

*L30 on P4, L17 you say you split the glaciers into three categories cracks, crevasses, scarps, here you say you are splitting them into two categories…*

On P4L17 we are talking about the type of surface disturbances. On P5L30 we are talking about the distinction in two separated categories of rate 3 rock glaciers. The reason why we create two categories of rate 3 rock glaciers is because most of previously known cases of destabilization report the occurrence of both crevasses and scarps (next sentence in text).

*P6L18 What predictor variable represents climatic characteristics?*

The elevation, as proxy of air temperature, and (partially) the PISR.

*P8L15 evidence*

Not sure what is meant here by the reviewer. We changed to "*evidence of permafrost occurrenc*e"

*P9L25 suggest .. m a.s.l, and slope angles ranging….*

Suggestion accepted

*L28 PISR*

Corrected

*L28 Based on the figure there doesn't appear to be an increase in destabilization predisposition around 2000 kWh/m2. Please explain.*

Thank you for noticing. This is a leftover from the previous version of the model before the major revision. Now the sentence has been deleted.

*L30 How did you explore the relationship between PTP and destabilization?*

It is now clarified at P9L26-28:

*"The relation between PTP and destabilization was also explored by including this predictor variable in the model instead of elevation. Although the PTP caused lower model performance, it could be observed that the PTP was positively correlated with the destabilization."*

*P10L1 The susceptibility map models rock glacier stability which you are using as a proxy to identify permafrost areas susceptible to destabilization correct?*

Yes correct.

*L5 This statement should be followed by some reasoning as to why "The susceptibility map predicted high destabilization susceptibility in areas belonging to stable rock glaciers"*

This is defined in the discussion (section 4.3, P12L17) , as in this section are reported only the results.

*P12L8 How was PTP found to be a significant predictor of potential destabilization? It wasn't included in the model. What statistics were done to measure significance?*

We now avoid to talk about significance. It is made clear that the relation between PTP and rock glacier stability is explored by a separate model where PTP is used instead of elevation (P9L26, see comment above)

*Where the map didn't perform well should be discussed. In the results you say "The susceptibility map predicted high destabilization susceptibility in areas belonging to stable rock glaciers" Why?*

This is discussed in the dedicated section 4.3

*Figure 1*

*Caption - Identification of the study area.*

Corrected

*In the legend what does "active production areas are found" mean? It would be useful to include another inset in the bottom left corner with a zoomed in image of one of the mountain ranges. It is difficult to see how the PFI scales with elevation on the map as it is.*

The legend is the one of the PFI map, it is now specified the reference of the PFI. This map refers to permafrost conditions at production areas of active rock glaciers. A zoom of the map has been inserted.

*Figure 2*

*What years are the orthoimages that are presented? It would be good if you actually included what you would have mapped on each orthoimage as it is quite difficult to see the features.*

The years are added (all 2013). A polygon in each image has been added showing what was actually mapped on these orthoimages.

*Figure 7*

*y-axis is very difficult to see. I would make it clear that you are presenting all variables however PTP wasn't included in the final model.*

Axes label fonts have been increased. It is now made clear in the caption that the PTP is not included in the final model.

*Figure 8*

*Is there a full susceptibility map for the entire study area? This may be of interest to many and should be included as a supplementary figure. Even better would be to include it in the paper and then have figure 8 as an inset.*

Yes there is a map of the region and is provided as supplementary material. We prefer not presenting it as figure here as it would be almost impossible to see the susceptible zones in such figure (see for example your comment on the figure 1).

*Tables*

*Table 4 Destabilization typo in column 1*

Corrected

*Table 5 Not called DEFROST susceptibility anymore right? Also based on the low surface area in the 3a class does it make sense to combine 3a and 3b into one class? Also please reorder 1, 2, 3a, 3b*

Thank you, DEFROST deleted. Classes are separated for consistency. Rates reordered, also in previous tables.

**Evaluating the destabilization susceptibility of active rock glaciers in the French Alps**

Marco Marcer[1,2], Charlie Serrano[1,2], Alexander Brenning[3], Xavier Bodin[2], Jason Goetz[3], and Philippe Schoeneich[1]

[1]Institut d'Urbanisme et Géographie Alpine, Université Grenoble Alpes, Grenoble, France
[2]Laboratoire EDYTEM, Centre National de la Recherche Scientifique, Université Savoie Mont Blanc, Le Bourget-du-Lac, France
[3]Department of Geography, Friedrich Schiller University Jena, Jena, Germany

**Correspondence:** Marco Marcer (marco.marcer@univ-grenoble-alpes.fr)

**Abstract.** In  this study, we propose a methodology to estimate the spatial distribution of destabilizing rock glaciers,  with a focus on the French Alps.  We mapped geomorphological features that can be typically found in cases of rock glacier destabilization (e.g. crevasses and scarps)  using orthoimages taken from 2000 to 2013. A destabilization rating was assigned  taking into account the evolution of  these mapped destabilization geomorphological features, and by observing the surface deformation patterns of the rock glacier,  also using the available orthoimages.  This destabilization rating served then as  to model the occurrence of rock glacier destabilization in relation to terrain attributes, and to spatially predict the susceptibility to destabilization at  a regional scale. Significant evidence of destabilization could be observed in 46 rock glaciers, i.e. 10% of the total active rock glaciers in the region.  Based on our susceptibility model of destabilization occurrence, it was found that this phenomenon is more likely to occur in elevations around the 0°C isotherm (2700 – 2900 m.s.l.), on  north-facing slopes, steep terrain (25° to 30°) and flat to slightly convex topographies. Model performance  was good (AUROC  = 0.76), and the susceptibility map  also performed well at reproducing observable patterns of destabilization. About 3 km$^2$ of creeping permafrost,  or 10 % of the surface occupied by active rock glaciers, had a high susceptibility to destabilization. Considering we observed that only half of  these areas of creep are currently showing destabilization evidence, we  suspect there is a high potential for future rock glacier destabilization within the French Alps.

**1 Introduction**

Warmer mean annual air temperatures (IPCC , 2013) are linked to a general trend of increasing permafrost temperature (e.g. Harris et al., 2003) and its water content (e.g. Ikeda et al., 2008) causing permafrost degradation, a phenomenon widely observed in the European Alps (Haeberli et al., 1993, 2010; Springman et al., 2013; Bodin et al., 2015).

 The occurrence of permafrost degradation is dependent on the ground properties, snow cover  and permafrost ice content (Scherler et al., 2013) and is therefore an heterogeneous phenomenon. Permafrost grounds affected by degradation experience a loss in  strength due to the increasing ice ductility and reduced internal friction caused by the warmer ice and increasing water content (Davies et al. , 2001; Haeberli et al., 1997; Harris and Davies , 2001; Nater et al., 2008; Huggel et al., 2010). Abnormal rockfall activity at high elevations (e.g. Ravanel and Deline , 2010) and increasing rock glaciers displacement rates (Delaloye et al., 2008) are often assumed as indicators of this change of state in the mountain permafrost. These processes may trigger mass movements that, in specific topographic conditions, may represent  a hazard to alpine communities. Therefore, there is a growing need to understand the occurrence of these phenomena at a regional scale to allow for a targeted risk assessment and land use planning (Haeberli et al., 2010).

In this context, rock glaciers experiencing destabilization  have recently become of interest. While active rock glaciers commonly present moderate interannual velocity variations that correlate with the ground temperature (Delaloye et al., 2008; Kellerer-Pirklbauer and Kaufmann , 2012; Bodin et al., 2009), destabilized rock glaciers are characterized by a significant acceleration that can bring the landform, or a part of it, to abnormally high velocities (Delaloye et al., 2013; Roer et al., 2008; Scotti et al., 2016; Lambiel, 2011; Eriksen et al., 2018). During this acceleration phase, morphological features typical of sliding processes, such as crevasses and scarps, appear and grow on the rock glacier surface. This suggests that the destabilization  occurrence is caused by a basal sliding process over the normal creep  (Roer et al., 2008; Schoeneich et al., 2015). In this sense, crevasses and scarps are interpreted as the possible transition between  creep-driven  sections and sliding sections of the landform (Roer et al., 2008). This  destabilization phase, also referred as a "surge" (Schoeneich et al., 2015) or a "crisis" (Delaloye et al., 2013), may last decades and it  usually results in a deceleration or inactivation of the landform. In very rare circumstances, destabilized rock glaciers may reach complete failure and collapse in a landslide (Bodin et al., 2016).

The destabilization process can be triggered either by a mechanical forces or changes in climate. An overload on the glacier surface caused by a landslide or glacio-isostatic uplift can cause a compressive wave that propagates through the landform increasing its displacement rates  leading to destabilization (Delaloye et al., 2013; Roer et al., 2008).  A warmer climate may also trigger a destabilization crisis as increasing temperatures may cause permafrost degradation of the rock glacier. This process may result in the onset of water saturated shear layers where sliding  occurs, triggering the crisis (Lambiel, 2011; Schoeneich et al., 2015; Eriksen et al., 2018). The onset of crevasses and scarps can also increase the predisposition of the landform to trap  water percolating into the permafrost body, causing a positive feedback process of destabilization (Ikeda et al., 2008). Although triggers are necessary to the destabilization occurrence, not all rock glaciers  subjected to these external forces destabilize. For example, permafrost degradation in rock glaciers mainly causes permafrost thaw  and results in inactivation (Scapozza et al., 2010). Destabilization  can be triggered only if there is a local topographical predisposition of the rock glacier to this process, such as steep slopes  (Roer et al., 2008; Delaloye et al., 2013). Therefore, the terrain attributes of the rock glaciers to the onset of a destabilization phase are a critical parameter in the process occurrence.

The purpose of this study was to obtain regional-scale insights into the issue of destabilizing rock glaciers in the French Alps.  Destabilization has been observed by several studies in the region (Echelard , 2014; Bodin et al., 2016; Serrano , 2017; Schoeneich et al., 20 ; however, there has not yet been a comprehensive assessment of this phenomenon. This was done by (i)  identifying the rock glaciers showing evidence of destabilization, in order to provide an assessment of destabilized landforms, and by (ii)  modeling the occurrence of this  phenomenon, in order to spot rock glaciers susceptible to incoming destabilization. Destabilized rock glaciers identification was performed by multi-temporal aerial image interpretation based on expert field knowledge (Section 2.2).  The geomorphological features typically occurring on destabilized landforms such as scarps and crevasses, here called "surface disturbances", were mapped and used to assign a destabilization rating ranging from 0 to 3 to each active rock glacier (Section 2.2.1). Rock glaciers  attributed with higher destabilization rating  have typical geomorphological charac-teristics reported in known cases of destabilization,  including pronounced surface disturbances that increased by number and size in the past decades. These rock glaciers were suggested to be potentially destabilized while  rock glaciers not presenting surface disturbances were classified with lower ratings of destabilization (i.e. stable rock glaciers).

The following step, i.e. modeling the destabilization occurrence, was performed by using a statistical approach  (that has been used for mapping landslide susceptibility (Goetz et al. (2011); Section 2.3).  Potentially destabilized rock glaciers were used as destabilization evidence and their relation with terrain attributes (e.g., slope angle and elevation) was modeled using a Generalized Additive Model (GAM). This model can be applied to better understand the  relation between destabilization occurrence and terrain predisposition, and to compute a destabilization susceptibility map, which provides an overview of potentially destabilizing landforms at  a regional scale (Section 2.3.1). Strengths and limitations of the methodology are widely discussed in the manuscript, as well as the contribution of the study to  enhancing our knowledge rock glacier destabilization.

**2 Methods**

**2.1 Study area and rock glacier inventory**

The French Alps cover an area approximately 50-75 km wide and 250 km long, located between 44° and 46°N and 5.7° to 7.7°W (Figure 1). Apart from the noticeably high Mont Blanc massif (peaking at 4810 m.a.s.l.), mountain ranges commonly peak between 3000 and 4000 m.a.s.l.. The lithology is heterogeneous across the region. The Northern French Alps can be roughly divided into the West side, dominated by granite and gneiss (ranges of Mont Blanc, Belledonne, Ecrins and Grandes Rousses), and East side, where ophiolites and schists are more common (ranges of Vanoise, Thabor and Mont Cenis). In the Southern French Alps ophiolites, limestone and mica schists are the most common lithology (ranges of the Ubaye), while the crystalline range of Mercantour can be found at the southernmost end of the region. Dominant geology is described the BRGM (2015) at 1/ 1 000 000 scale, and the vectorial version of this map is used in this study to observe destabilization occurrence in relation to lithology.

In this region permafrost was estimated to cover up to 770 km$^2$ (Boeckli et al., 2012; Marcer et al., 2017). The 0°C annual isotherm at the end of the $20^{th}$ century ranged from 2500 m a.s.l. in the south to 2300 m a.s.l. in the north (Gottardi , 2009). The periglacial landforms of the region were inventoried by the "Office national des forêts" (ONF: the National Forest Office) (Roudnitska et al., 2016), and revealed the high presence of active rock glacier in the region (i.e. 493 mapped rock glaciers). This inventory was compiled between the years 2009 – 2016 by inspecting aerial imagery and revised by Marcer et al. (2017). This inventory was used in the present study to identify active rock glaciers locations and to investigate the occurrence of destabilization.

According to Auer et al. (2007), mean annual air temperature increased by up to 1.4°C in the French Alps during the $20^{th}$ century, and this rate has been increasing in recent decades. This climate warming is suspected to have caused some noticeable effects on the permafrost characteristics in the region. The only deep permafrost borehole in the region, located in the Ecrins massif in temperate permafrost (-1.3°C) with low ice content, showed a temperature increase rate of 0.04°C per decade between 2010 and 2014 (Schoeneich et al., 2012), similar to others sites in Switzerland where data series are longer (PERMOS , 2016). Increasing air temperature was also addressed to be responsible for the acceleration since the late 1990s of the active Laurichard rock glacier located in the Combeynot massif of the French Alps (Bodin et al., 2009). Several cases of rock glacier destabilization were also observed in the region, as the collapsed Berard rock glacier (Bodin et al., 2016) and the Pierre Brune rock glacier (Echelard , 2014). Serrano (2017)

mapped destabilized rock glaciers in the Maurienne valley, Vanoise national park and Ubaye valley, highlighting the high incidence of destabilized rock glaciers in these areas.

**2.2 Mapping rock glacier destabilization**

The first step to identify destabilized rock glaciers was mapping surface disturbances on rock glaciers. Previous studies that described destabilized rock glaciers showed that these landforms present a wide variety of geomorphological features (e.g. Roer et al., 2008). Here, we followed a methodology similar to Serrano (2017), which consisted of defining a catalogue of typical surface disturbances that can be found on destabilized rock glaciers. Surface disturbances on rock glaciers were classified in three distinct categories, depending on their morphology: cracks, crevasses and scarps. Surface disturbances are described in detail in Table 1 and illustrated in Figure 2.

In this study, surface disturbances were mapped for the inventoried rock glaciers based on interpretation of a set of multi-temporal high-resolution aerial imagery for the French Alps. This orthoimagery collection was obtained from the Institut géographique national (IGN, National Institute of Geography), which is freely available from the official website (www.geoportail.fr) or can be accessed as a Web Mapping Service (IGN , 2011a, 2013). The IGN orthoimagery collection consists of orthomosaics covering all of France for three different collection periods. The first orthomosaic is composed of images taken from 2000 to 2004, the second from 2008 to 2009, and the third from 2012 to 2013. All images are of high-resolution: 50 cm x 50 cm for the most recent mosaic and slightly lower values (1 m x 1 m at its lowest) for the older mosaics, depending on the location. This resolution was sufficient to identify the smallest features to be mapped, i.e. the surface cracks (Figure 2a). Nevertheless, several limitations during the mapping process were encountered, as image distortion or illumination, and will be discussed in section 4.4.1.

Using a single orthoimage to map surface disturbances can lead to misinterpretations in the case of poor illumination of the terrain and snow patches covering the ground (Serrano , 2017). Indeed, as the surface morphology of a rock glacier is naturally shaped according to spatially varying creep patterns, it is easy to mistake actual surface disturbances related to compression features, such as furrows, depending on image quality. Therefore, surface disturbances, i.e. those morphological features not related to the creeping of the ice-rich permafrost, were mapped using all three available orthoimages in order to check that actual strain occurred where surface disturbances  were located and to overcome limitations related to poor quality of an individual image.

**2.2.1 Rating the degree of destabilization**

After the rock glacier surface disturbances were mapped, a rating of the degree of destabilization was assigned to each rock glacier. This rating was given not only to provide some insight to the observed levels of destabilization in the French Alps, but also to provide a confidence rating to describe a rock glacier as stable or unstable for the spatial distribution modelling of rock glacier destabilization.

Assigning a rating to quantify the degree of destabilization of a rock glacier required the definition of the characteristics of the "typical" destabilized rock glacier that can be observed on multiple orthoimages. To do so, we investigated the

 features of destabilized rock glaciers reported in the literature that could be observed by orthoimagery interpretation.  At first, it was observed that the presence of surface

5 disturbances was a necessary but not sufficient condition to the occurrence of destabilization, as rock glaciers may present surface disturbances but be stable for decades. For example, in the Pierre Brune, Roc Noir  and Hinteres Langtalkar rock glaciers, although crevasses could be observed in aerial imagery since the 1940s to the 1960s, destabilization occurred only in the late 1990s

10 (Echelard , 2014; Serrano , 2017; Roer et al., 2008). Second, the destabilization process can be linked to an increase of surface disturbances occurrence (see Figure 3). Also, surface disturbances on destabilized landforms were observed to create a discontinuity in the creep pattern. For example, the Plator, Grosse Grabe and Gänder rock glaciers

15  have gone through a sharp transition from displacement speeds in the order of 0.1 – 0.9 m/y to displacements speeds of the order of several meters per year (Scotti et al., 2016; Delaloye et al., 2008).

 Finally, a high displacement rate may not be a necessary feature, as some destabilized rock glaciers, e.g. Lou and

20 Furggwanghorn, moved at a "normal" rate of around 2 m/yr (Schoeneich et al., 2017; Roer et al., 2008).

These observations suggest that destabilization may be spotted in orthoimages if the landform has surface disturbances increasing overtime time by frequency and/or magnitude, as well as if disturbances also create a strong discontinuity in the deformation pattern of the

25  landform. Nevertheless, rock glaciers were observed to show a wide variety and combination of these features, making it unrealistic to construct a binary classification of stable versus destabilized landforms. In order to acknowledge this, we proposed a rock glacier  destabilization rating based on four rates that varied from 0 (stable rock glaciers) to 3 (rock glaciers potentially destabilized),  which is explained in more detail in Table 2. For each active rock glacier, a rating of

30 the degree of destabilization was assigned by observing the combination of surface disturbances and a qualitative assessment of recent deformation patterns. This rating was applied using a standardized workflow (Figure 4). A comparison of the available IGN multi-year orthoimagery was used to observe the temporal evolution of the surface disturbances and surface deformation patterns.

[revised manuscript text omitted]

**2.3.2 Model predictor variables**

Terrain attributes used in modelling needed to be selected to act as proxies for processes that precondition destabilization. Although destabilization is found to occur in different conditions, some topographical features seem to

be recurrent. Destabilization  has been observed to occur on steep slopes, as high slope angles tend to increase the internal shear stress (Delaloye et al., 2013). Surface disturbances are often located in convex  shaped bedrock surfaces, which causes an extensive flow pattern and a thinning of the permafrost body (Delaloye et al., 2013). Solar exposure also may be significant in the destabilization occurrence  since all known cases of destabilized rock glaciers in the French
5 Alps are  north facing. Solar exposure can also be a proxy of the snow cover duration, as  north-facing slopes are more prone to conserve longer snow patches through the summer  making melt water available through the summer. Elevation  and mean annual air temperature  can also be proxies of snow cover duration that have the possibility to affect permafrost characteristics. Considering this, slope angle, profile curvature, potential incoming solar radiation (PISR) and elevation were tested as predictor variables.

10 Terrain attributes were derived from the BD Alti DEM, 25 m × 25 m spatial resolution (IGN , 2011a). Slope angle and downslope curvature (Freeman , 1991) were evaluated using the Morphometry Toolbox in SAGA GIS (version 2.2.2, Conrad et al. 2015). Negative values of curvature indicate concave topography, while positive values indicate convex topography. Also PISR was calculated using the Terrain analysis toolbox in SAGA as the sum of the computed direct and diffusive components of the radiation (Wilson and Gallant , 2000). Clear-sky conditions, a transmittance of 70 %, and absence of a snow cover were
15 assumed in the calculation of the annual total PISR.

Finally, it was decided to evaluate the relation between rock glacier destabilization and the spatial distribution of degrading permafrost in order to give an insight on the significance of the warming climate with respect to the destabilization phenomena. The spatial distribution of degrading permafrost was evaluated following the method already presented by other studies (Hoelzle and Haeberli , 1995; Lambiel and Reynard , 2001; Damm and Felder , 2013), which consisted  of artificially shifting a
20 permafrost map proportionally to the estimated climate warming  occurring between the period of validity of the map and the current climate. Here,  we used a Permafrost Favourability Index (PFI) map (Marcer et al., 2017)  to acts as a permafrost distribution map for the region. The PFI map was calibrated using active rock glaciers as  evidence of permafrost occurrence, and it represents the permafrost conditions during the cold episodes of the Holocene, e.g. Little Ice Age (LIA). The climate warming between the years 1850-1920
25 and 1995-2005 was determined using the HISTALP database (Auer et al., 2007) over the region. A permafrost distribution map was then recomputed taking into account  these temperature variations and represented the theoretical permafrost distribution in equilibrium with the current climate. By comparing this theoretical permafrost distribution and the PFI,  a map of the Potential Thawing Permafrost zone (PTP, i.e. the so-called "melting area" in Lambiel and Reynard  (2001)) was obtained. In order to use the PTP as predictor variable, it was represented by an index ranging between 0, i.e. no thaw expected,
30 and 1, i.e. potential thaw.

It  should be emphasized that PTP is only a proxy of permafrost degradation, which occurs at all the elevations, while the PTP zone consists  of a belt of 250 to 300 meters  in elevation that affects about 50% of the lower margins of the permafrost zone (Figure 5). PTP is used under the hypothesis that degradation is more intense at the lower margins of the permafrost zone  where permafrost conditions may be more temperate, richer in water, and more sensitive
35 to climate variations.

**2.3.3 Susceptibility modelling**

The model  of rock glacier stability was also used to predict the occurrence of degrading permafrost over the French Alps  by producing a susceptibility map (e.g. Goetz et al., 2011). This was done using the R package RSAGA and the raster images of the predictor variables maps, which allowed  extrapolating the relationships between rock glacier stability and terrain attributes at the landscape scale.  We would like to highlight that since the model is constructed using data on destabilized rock glaciers, the susceptibility map  applies mainly for processes relative to destabilization of ice-rich debris slopes. Therefore, in areas where creeping permafrost does not exist, the extrapolated susceptibility may have high uncertainty. The model predicted a DEFROST index which was classified into five susceptibility zones using the 50, 75, 90, and 95 percentiles (Rudy et al., 2017; Goetz et al., 2011). These zones described very low (<50), low (50 – 75), medium (75 – 90), high (90-95) and very high (>95) susceptibility to permafrost destabilization.

**3 Results**

**3.1 Destabilized rock glaciers inventory**

More than 1300 surface disturbances were digitized, involving 259 active rock glaciers (Figure 6). Overall, more than the 50% of the active rock glaciers may be affected by some degree of destabilization as 46 rock glaciers (9.7%) showed potential destabilization, 86 (17.0%) were suspected of destabilization and 127 (25.7%) were unlikely destabilized. Only 13 potentially destabilized rock glaciers presented deep surface disturbances.

Potentially destabilized rock glaciers were mainly located in in the Vanoise National Park and in the Queyras and Ubaye mountain ranges. In these areas, densely jointed lithologies  (i.e., ophiolites and schists ) dominate. Rock glaciers in crystalline lithologies  (i.e., gneiss and granite ) were found to have low rates of destabilization. That is, only two rock glaciers were rated as possibly destabilized over a population of 55 (Table 3).

The predominant surface disturbance observed were cracks, which were present  in 187 of the active rock glaciers (Table 4). Crack clusters also had a high number of observed cases (152), while the deep surface disturbances occurred in about 15% of all the examined rock glaciers. In general, the  occurrences of surface disturbances were dependent on the destabilization rating. Scarps and crevasses were found in about 10% on unlikely destabilized landforms. The observation of each surface disturbance was highest for potentially destabilized rock glaciers with deep surface disturbances, indicating that in these landforms multiple surface disturbances coexist.

**3.2 Modelling**

Following a stepwise backward and forward selection, the chosen model included PISR, slope angle, elevation and curvature as predictors. The mean cross-validated AUROC was 0.76 on the test set, indicating a good performance (Hosmer and Lemeshow

, 2000). The predictors having most influence on the response variable were the PISR (AUROC change = 0.162), curvature (AUROC change = 0.068), slope angle (AUROC change = 0.031) and elevation (AUROC change = 0.018).

The model transformation functions revealed the relations between terrain attributes and rock glacier stability (Figure 7).  Higher predisposition to destabilization was more likely to occur in an altitudinal range between 2700 and 2900 m a.s.l.  and slope angles ranging between 25 and 30°. Slightly negative to positive curvature was also favourable to destabilization. PISR was negatively correlated with the destabilization probability, indicating that rock glacier destabilization was more likely to occur on north-facing slopes.  The relation between PTP and destabilization was also explored by including this predictor variable in the model instead of elevation. Although the PTP caused lower model performance, it could be observed that the PTP was positively correlated with the destabilization.

**3.3 Susceptibility map**

[revised manuscript text omitted]

Modeling rock glacier destabilization using PTP instead of elevation revealed that an increasing potential in permafrost thaw was linked to an increase in susceptibility to destabilization, indicating that destabilization was more likely to occur where the permafrost zone was expected to be thawing. This seems to be consistent with the relationship between destabilization and elevation, as potentially destabilized rock glaciers are more often located around 2800 m.a.s.l., which roughly coincides with the lower margins of the regional permafrost zone.

**4.3 Susceptibility map**

Overall, permafrost destabilization was adequately described, as indicated by the cross-validated performance, in most of the observed cases of destabilization. Although cases of potential destabilization were inventoried, rock glaciers that have a low rating of destabilization and are located in areas with high susceptibility should be identified as having a high potential of  future destabilization. Results indicated that these rock glaciers had a large area of high predisposition to destabilization, suggesting that there is a high potential for future destabilization in the region. The map therefore may be used to spot rock glaciers that present a predisposition to develop destabilization. In particular, the Laurichard rock glacier is a site currently under monitoring which was found to present a low to medium susceptibility to destabilization in this study (Bodin et al., 2008). The comparison of the future evolution of this landform with respect to the  susceptibility map is therefore recommended.

**5 Conclusions**

The present study aimed to give insights into the extent of  destabilizing rock glaciers in the French Alps.  Mapping and modelling rock glacier destabilization in  this region was conducted using an orthoimagery collection, a 25 
[revised manuscript text omitted]

|  Destabilization rating | Cracks | Crack clusters | Crevasses | Scarps |
|---|---|---|---|---|
| **1** | 86 | 54 | 13 | 8 |
| **2** | 52 | 51 | 15 | 11 |
| ** 3a** | 10 | 9 | 10 | 8 |
| **3b** | 23 | 29 | 0 | 0 |
| **Totals** | **187** | **152** | **40** | **27** |

**Table 5.** Active rock glacier area per class of destabilization susceptibility.

| Destabilization Rating | Surface per susceptibility class [km$^2$] | | | | |
|:---:|:---:|:---:|:---:|:---:|:---:|
| | *Very Low* | *Low* | *Medium* | *High* | *Very High* |
| **0** | 8.09 | 3.21 | 1.70 | 0.43 | 0.37 |
| **1** | 4.03 | 2.16 | 1.29 | 0.42 | 0.38 |
| **2** | 2.18 | 1.50 | 0.93 | 0.34 | 0.30 |
|  **3a** | 0.17 | 0.27 | 0.17 | 0.05 | 0.05 |
| **3b** | 0.07 | 0.19 | 0.31 | 0.24 | 0.38 |
| **Cumulative Surface** | 14.54 | 7.33 | 4.41 | 1.47 | 1.48 |